

# Pathway analysis identifies altered mitochondrial metabolism, neurotransmission, structural pathways and complement cascade in retina/RPE/choroid in chick model of form-deprivation myopia

Loretta Giummarra[1], Sheila G. Crewther[1], Nina Riddell[1], Melanie J. Murphy[1] and David P. Crewther[2]

[1] School of Psychology & Public Health, La Trobe University, Melbourne, Victoria, Australia
[2] Centre for Psychopharmacology, Swinburne University of Technology, Hawthorn, Victoria, Australia

## ABSTRACT

**Purpose**. RNA sequencing analysis has demonstrated bidirectional changes in metabolism, structural and immune pathways during early induction of defocus induced myopia. Thus, the aim of this study was to investigate whether similar gene pathways are also related to the more excessive axial growth, ultrastructural and elemental microanalytic changes seen during the induction and recovery from form-deprivation myopia (FDM) in chicks and predicted by the RIDE model of myopia.

**Methods**. Archived genomic transcriptome data from the first three days of induction of monocularly occluded form deprived myopia (FDMI) in chicks was obtained from the GEO database (accession # GSE6543) while data from chicks monocularly occluded for 10 days and then given up to 24 h of normal visual recovery (FDMR) were collected. Gene set enrichment analysis (GSEA) software was used to determine enriched pathways during the induction (FDMI) and recovery (FDMR) from FD. Curated gene-sets were obtained from open access sources.

**Results**. Clusters of significant changes in mitochondrial energy metabolism, neurotransmission, ion channel transport, G protein coupled receptor signalling, complement cascades and neuron structure and growth were identified during the 10 days of induction of profound myopia and were found to correlate well with change in axial dimensions. Bile acid and bile salt metabolism pathways (cholesterol/lipid metabolism and sodium channel activation) were significantly upregulated during the first 24 h of recovery from 10 days of FDM.

**Conclusions**. The gene pathways altered during induction of FDM are similar to those reported in defocus induced myopia and are established indicators of oxidative stress, osmoregulatory and associated structural changes. These findings are also consistent with the choroidal thinning, axial elongation and hyperosmotic ion distribution patterns across the retina and choroid previously reported in FDM and predicted by RIDE.

Corresponding author
Sheila G. Crewther,
s.crewther@latrobe.edu.au

## INTRODUCTION

Myopia (short-sightedness) is the most common visual disorder worldwide and the greatest risk factor for many severe ophthalmic diseases in older individuals (*Dolgin, 2015*). Rapidly increasing prevalence has been reported among young adults in areas of South East Asia (*Saw et al., 2000*) since the 1970s, implicating environmental influences such as changing lifestyles and education as key factors in myopia development (*Dolgin, 2015*; *Junghans & Crewther, 2003*; *Morgan, Ohno-Matsui & Saw, 2012*; *Schneider et al., 2010*).

Many of the morphological and physiological characteristics seen in clinical myopia are associated with conditions such as macular oedema, age related maculopathy (AMD), retinal detachment, glaucoma and choroidal neovascularisation (CNV) (*Seet et al., 2001*; *Yap, Cho & Woo, 1990*). In particular, the elongation of the vitreal chamber, ocular volume increase, thinning of the retina and choroid and reduced choroidal blood flow in clinical myopia (*Borish, 1949*; *Morgan, Ohno-Matsui & Saw, 2012*; *Moriyama et al., 2007*; *Yang & Koh, 2015*; *Zhang & Wildsoet, 2015*) implicate mechanisms associated with impaired transport of fluid from vitreous to choroid as contributors to significantly greater physiological risk of loss of vision and blindness.

Experimental models of myopia came to prominence in the late 70s in monocularly occluded monkeys (*Raviola & Wiesel, 1978*) and chickens (*Wallman, Turkel & Trachtman, 1978*). In animal models, particularly chicken, form-deprivation (FD) is characterised by rapid ocular growth and development of myopia via dramatic increases in vitreous chamber volume (*Wallman, Turkel & Trachtman, 1978*), reduced choroidal blood flow and concurrent choroidal and retinal thinning (*Shih et al., 1993*; *Shih, Fitzgerald & Reiner, 1993a*; *Shih, Fitzgerald & Reiner, 1993b*) (and as shown with the MRI photomicrograph in Fig. 1), similar to that seen in profound human myopia (*Borish, 1949*; *Feldman et al., 1991*; *Morgan, Ohno-Matsui & Saw, 2012*; *Moriyama et al., 2007*; *Yang & Koh, 2015*; *Zhang & Wildsoet, 2015*).

Similarity in observations of human and animal models of myopia led to the formulation of the Retinal Ion Driven Efflux (RIDE) model of myopia (*Crewther, 2000*). This theory proposes that acute blur will perturb the rate of exchange of ions and fluid between photoreceptors and the sub-retinal space, concurrently affecting neurotransmission (*Westbrook, Crewther & Crewther, 1999*), tissue osmoregulation (*Crewther et al., 2006*) and metabolic pathways across the posterior eye (*Riddell et al., 2016*). As a consequence, inhibition of normal efflux of fluid across the retina/RPE would result in increases in vitreous volume, axial growth and induce changes in refractive status leading to myopia.

The relevance of the chick model in particular to understanding of human myopia has recently been highlighted by *Riddell & Crewther (2017a)* who first demonstrated that the genes near human GWAS of myopia identified refractive error loci that significantly overlap with the genes differentially expressed in animal transcriptome studies. Furthermore,
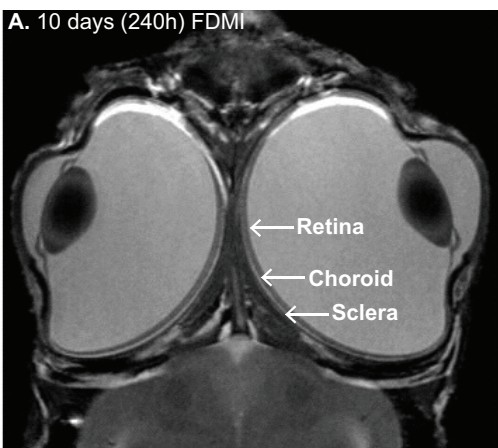 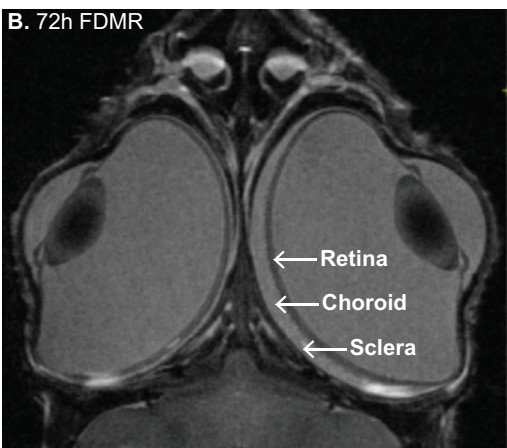

**Figure 1** **MRI images of chick eye following 10 days of FDM induction and 3 days recovery.** (A) Monocular form-deprivation of the right eye (RE) for 10 days demonstrates abnormal ocular growth, excess vitreal volume, and thinned choroid of RE compared to its fellow left eye (LE). (B) 72 h post-occlusion recovery (i.e., 72 h normal visual experience) resulted in vitreous volume decrease and choroidal expansion. Previous studies (*Liang et al., 2004*) have shown ~300% increase in choroidal thickness in RE compared to LE after three days recovery from FD. Note: Images (same magnification) in (B) slightly more dorsal than in (A). Image credit: G Egan.

*Riddell & Crewther (2017b)* also showed that the genes and proteins differentially expressed in chick myopia and hyperopia models overlap significantly with the genes and proteins implicated in the pathogenesis of sight-threatening secondary disorders.

Previous ultrastructural studies of form-deprivation in chick have demonstrated morphological abnormalities in photoreceptor outer segments, retinal pigment epithelium (RPE) nuclei, mitochondria and basal laminae (*Beresford, Crewther & Crewther, 1998*; *Liang et al., 1995*; *Liang et al., 2004*; *Liang et al., 1996*) similar to those described elsewhere as characteristic of AMD (*Datta et al., 2017*), CNV (*Ohno-Matsui et al., 2017*) and glaucoma (*Kim & Park, 2017*). These morphological changes occur concomitantly with elemental microanalytical evidence of hyperosmotic changes in ion distribution patterns across the retina, RPE and choroid (*Crewther et al., 2006*; *Grubman et al., 2016*; *Hollborn et al., 2017*; *Junghans et al., 1999*; *Liang et al., 1995*; *Liang et al., 2004*) and reminiscent of physiologically induced hyperosmotic and oxidative stress elsewhere in the brain (*Brocker, Thompson & Vasiliou, 2012*; *Morland, Pettersen & Hassel, 2016*; *Veltmann et al., 2016*). Indeed, oxidative stress has been suggested to contribute to the underlying mechanisms involved in profound myopia pathology (*Francisco, Salvador & Amparo, 2015*).

Our recent RNA sequencing analysis of the genomic changes associated with early optical induction of myopic and hyperopic refractive errors (*Riddell et al., 2016*) also suggested that metabolic pathways will be altered in any genomic analysis of environmentally induced change in light driven neurotransmission particularly in the most extreme of form-deprivation (FD).

Over the last decade there have been many large discovery type transcriptome studies examining the genomic basis of environmentally altered eye growth in animal models of refractive error development (*Ashby & Feldkaemper, 2009*; *Brand, Schaeffel & Feldkaemper, 2007*; *Guo et al., 2013*; *Guo et al., 2014*; *He et al., 2014*; *McGlinn et al., 2007*; *Rada & Wiechmann, 2009*; *Riddell et al., 2016*; *Schippert, Schaeffel & Feldkaemper, 2008*; *Schippert, Schaeffel & Feldkaemper, 2009*; *Shelton et al., 2008*; *Stone et al., 2011*; *Tkatchenko et al., 2006*). As with human genome-wide association studies (GWAS), the majority of the identified genes converge into biological pathways such as cell structure, cell–cell communication, neurotransmission, retinoic acid metabolism, ion transport, energy metabolism, immune system and eye development (*Hysi et al., 2014*; *Kiefer et al., 2013*; *Riddell & Crewther, 2017a*; *Stone & Khurana, 2010*; *Verhoeven et al., 2013*). However, the large number of genes implicated in both human GWAS and animal transcriptome studies have not offered a coherent explanation for the anatomically derived evidence of mitochondrial and hyperosmotic stress seen across the posterior retina/RPE/choroid of the FDM eye. Thus, the aim of this study was to investigate whether similar gene pathways are also related to the excessive axial growth, ultrastructural and elemental microanalytic changes seen during the induction and recovery from form-deprivation myopia (FDM) in chick.

To test the expected association between the ultrastructural changes and the RIDE model (*Crewther, 2000*) during the induction and recovery from FDM that we have previously examined (*Crewther et al., 2006*; *Liang et al., 2004*), we have reanalysed archived genomic data from *McGlinn et al. (2007)* and a new transcriptomic dataset from chicks with FDM. The previously published microarray study (*McGlinn et al., 2007*) analysed retina and RPE tissue at 6 h and 72 h of FD myopia induction. This dataset was obtained from the Gene Expression Omnibus database (accession number GSE6543). This dataset was reanalysed using Gene Set Enrichment Analysis (GSEA) method in conjunction with our novel FDMR data using retina/RPE/choroid tissue following 10 days of translucent occlusion, at time of occluder removal and then at 6 h and 24 h post occlude removal. The RIDE model would predict that pathways associated with neurotransmission, metabolism and ion solute transport would be significantly perturbed

Gene Set Enrichment Analysis (GSEA) (*Subramanian et al., 2005*) has been utilized to identify key expression networks involved in induction of form-deprivation myopia (FDMI) and during early recovery from form-deprivation myopia (FDMR). GSEA was originally developed to identify consistent generalized differences in the cumulative distribution in the expression of genes in a biological pathway based on a priori knowledge of the gene's biological function (*Subramanian et al., 2005*). Unlike other pathway analyses, all genes within the expression dataset are considered irrespective of whether particular DEGs show statistically significant differences in gene expression as identified in previous analyses.

## MATERIALS AND METHODS

### FD induction dataset

Previously published microarray data from the *McGlinn et al. (2007)* study were obtained from the GEO Database (https://www.ncbi.nlm.nih.gov/geo/; accession number GSE6543). The raw CEL files from this study were reanalysed to complement our FD recovery profile. Although many microarray studies have ascertained the transcriptome profile of refractive error development and its progression (*Ashby & Feldkaemper, 2009*; *Brand, Schaeffel & Feldkaemper, 2007*; *McGlinn et al., 2007*; *Rada & Wiechmann, 2009*; *Schippert, Schaeffel & Feldkaemper, 2008*; *Schippert, Schaeffel & Feldkaemper, 2009*; *Shelton et al., 2008*; *Stone & Khurana, 2010*; *Tkatchenko et al., 2006*), all have used optical defocus or FD in other species. The *McGlinn et al. (2007)* study is the most comparable to ours in that similar tissue was collected, i.e., chick Retina/RPE preparation, from FD animals and analysed using the same Affymetrix microarray chips. To accompany our re-analysis of this data, refraction and ocular biometrics were collected from 24 chickens that were monocularly occluded at one-week of age for 6 hr ($n = 6$) and 72 hr ($n = 6$) with an additional 12 chicks used as age-matched controls.

### FD recovery dataset
#### Animals

Twenty hatchling chicks (*Leghorn x New Hampshire*) were utilised in this study. Fifteen chicks were form-deprived (FD) for 10 days (day 2-11 post-birth) by attaching a translucent polystyrene occluder to the periocular feathers of their right eye as previously described (*Crewther et al., 2006*). Separate chicks were used as aged-matched unoccluded controls ($n = 5$). Occluders were removed on day 12 and chicks were given 0 hr ($n = 5$), 6 hr ($n = 5$) or 24 hr ($n = 5$) of normal vision to recover from form-deprivation. Chicks were raised with unlimited food and water in a controlled environment on a 12 h light/12 h dark cycle and with the temperature maintained at 30 ± 0.5 °C. Illuminance was maintained at 183 lux during the 12 h day cycle using a 20 W halogen lamp. All animal work in this study was approved by the La Trobe University Animal Ethics Committee (Approval No. 05/07) and is in accordance with the Guidelines for Use of Animals in Research by the National Health and Medical Research Council (NHMRC) of Australia and the ARVO Statement for the Use of Animals in Ophthalmic and Vision Research.

#### Ocular refraction, biometric analysis

Refractive state (dioptres (D)), vitreous chamber depth (VCD in mm) and axial length (AL in mm) measures were collected from all animals (induction and recovery) while animals were surgically anesthetized with an intramuscular injection of ketamine (45 mg/kg) and xylazine (4.5 mg/kg). Refraction in the experimental right eyes were determined by trained ophthalmic practitioners using retinoscopy (Keeler, Vista Diagnostic Instruments, Malvern, PA, USA) and A-Scan ultrasonography (A-Scan III, TSL; Teknar, Inc. St Louis, USA; 7 MHz probe) was used to measure axial dimensions. Analyses of Variance (ANOVA) was used to test group differences in refraction, AL and VCD followed by post-hoc tests
if required. All dependent variables met the assumption for equal variance (Levene's Test $p < 0.05$).

### MRI imaging

MRI (4.7T) of chick FD for 10 days then given 72 h of normal visual experience was obtained to confirm our previous histological analysis (*Liang et al., 2004*) which has shown ~300% increase in choroidal thickness in the FD eye compared to fellow eye. For this, chicks were stereotaxically immobilised under surgical anaesthesia in the small bore of the magnet. Chick heart rate was also monitored.

### Microarray tissue collection and RNA isolation

All chicks were euthanized by decapitation immediately after ocular measurements were taken. Right eyes were enucleated and the choroid/retina/RPE were taken from the posterior eye cup, placed in PrepProtect™ RNA stabilizing buffer (Miltenyi Biotec Australia Pty. Ltd., North Ryde, NSW, Australia) and stored on ice until transferred to −20 °C freezer. RNA was extracted using the SV total RNA isolation system (Promega Australia, NSW, Australia), including DNA digestion. The quality of the RNA samples was assessed via NanoDrop® ND-1000 Spectrophotometer (ThermoFisher, Waltham, MA, USA) and found to fall within the acceptable absorbance (260/280) range of 1.8–2.1. For Affymetrix microarrays, RNA from the right-eyes of each animal (control and experimental) was pooled in equimolar amounts by experimental condition (control ($n = 5$), 0 hr ($n = 5$), 6 hr ($n = 5$) and 24 hr ($n = 5$)) and sent to the Australian Genome Research Facility Ltd (Walter and Eliza Hall Institute, Victoria, Australia) for microarray processing. Raw data were exported as CEL files containing probe level intensities for preprocessing with Expression Console™ 1.1 (Affymetrix, Inc, Santa Clara, CA, USA). This data has been submitted to GEO Database (https://www.ncbi.nlm.nih.gov/geo/; accession number GSE89325).

### Sample pooling

Pooling of RNA samples was chosen, as our primary aim was to identify altered biological pathways associated with experimental myopia using GSEA rather than single-gene analysis. GSEA assesses the collective changes in gene expression and identifies relevant biological pathways where these genes act (*Manoli et al., 2006*; *Subramanian et al., 2005*). Pooling biological samples was originally discouraged for single gene analysis as pooling may preclude variance measures in downstream statistical analysis (*Peng et al., 2003*). Many later publications indicate that such caution is unnecessary as sample pooling does not impact negatively on identifying differentially expressed genes, particularly for small experiment designs and in animals within similar experimental manipulations (*Bottje et al., 2012*; *Fu et al., 2011*; *Mengozzi et al., 2012*; *Mustafi et al., 2011*; *Zhang et al., 2007*). Furthermore, such limitations in sample pooling does not have impact on the reliability of the GSEA algorithm which requires at least a subset of genes within a pathway to be consistently ranked near the top or bottom of the ordered list rather than if single gene measures were used (*Manoli et al., 2006*; *Subramanian et al., 2005*). In addition, to ensure the rigor of our analyses we used a more stringent statistical threshold of 0.05 for our

analysis rather than the recommended 0.25 for GSEA (*Manoli et al., 2006*; *Subramanian et al., 2005*).

### Data pre-processing & normalisation

To determine if the differences in chicken strain and tissues used produced confounding results, we pre-processed the GSE6543 and our Affymetrix chicken chip data (CEL files) individually and then together using Expression Console™ 1.1 (Affymetrix, Inc, Santa Clara, CA, USA). No significant outliers were found in either forms of pre-processing therefore we chose to present the data that was modelled together. Average background, RawQ, poly-A controls (*dap, lys, phe, thr, trp*) & hybridisation controls (*bioD, bioC, bioD, cre*) were assessed (*Affymetrix, 2004*; *Affymetrix, 2006*). The raw data was summarised and normalised using the Robust Multichip Average (RMA) algorithm to yield log base 2 expression values for each transcript. Expression values of genes with multiple probe sets were then median summarised, resulting in a total of 14,298 gene measures. Expression values of the FD induction samples were averaged by condition (6hr controls ($n = 1$), 6 hr FD ($n = 1$), 72 hr control ($n = 1$), 72 hr FD ($n = 1$)) to yield a single log2 expression value. This was done as FDMR tissue samples were pooled prior to microarray analysis and hence resulting in one sample (ie data point) per time-point.

Although all (i.e., induction and recovery) CEL files were modelled together, average background scores ranged from 56 to 114 for the FD induction data and from 62 to 89 for the FDMR dataset. These values are mostly consistent with Affymetrix recommendations of typical average background values falling between 20 to 100 (*Affymetrix, 2004*). Poly-A controls were all present with average signal of *dap>thr>phe>lys*. Hybridisation controls were also present with increasing signals which reflect their relative concentrations, specifically *cre>bioD>bioC>bioB*. Ideally, arrays being compared should have comparable background values so these findings may have resulted from electrical noise rather than low sample quality as other parameters were consistent with manufacturer recommendations. Both GAPDH and EF1 $\alpha$ internal controls indicated that hybridisation fell within the parameters. Interestingly, the β-actin signal was above threshold for both datasets, possibly indicating that specific transcription of β-actin is altered in extreme myopia. Indeed, phototransduction requires actin filaments and microtubules to redistribute arrestin and transducin (*Reidel et al., 2008*) making β-actin an unreliable housekeeping gene (*De Boever et al., 2008*). Furthermore, our own ultrastructure studies have indicated redistribution of actin filaments in cells such as RPE and photoreceptors in form-deprivation (*Liang et al., 2004*). Thus, pre-processing of both datasets was rerun to mask the Affymetrix housekeeping probesets for β-actin. After median summarisation, the final dataset included 14,298 genes. A report of the parameters used in the pre-processing of both the GSE6543 and our Affymetrix chicken chip datasets is included in Table S1. Now available as GEO Database (https://www.ncbi.nlm.nih.gov/geo/; accession number GSE89325).

### Gene set enrichment analysis

The Broad Institute's Gene Set Enrichment Analysis (GSEA) software was used to determine whether a priori defined sets of genes were significantly enriched (*Mootha*

*et al., 2003*; *Subramanian et al., 2005*) during the induction and recovery of FD. Curated gene sets were obtained from the Molecular Signature Database (MSigDB). In particular, annotated gene sets were sourced from three databases; BioCarta (http://www.biocarta.com/genes/index.asp), KEGG (http://www.genome.jp/kegg/pathway.html), and Reactome (http://www.reactome.org/) and chick genes were converted to human genes where possible (Table S2). Pearson's correlation was used, being the recommended metric for time-series data (*Broad Institute MIoTM, 2012*) to assess changes in gene expression over the duration of occluder wear, both short-term (6 hr & 72 hr; GSE6543) and long-term FDMI which combined data from GSE6543 (retina/RPE; 6 hr & 72 hr) and GSE89325 (retina/RPE/choroid; 240 hr). Changes in gene expression was also assessed during the recovery after occluder removal (FDMR; 0 hr, 6 hr, & 24 hr). GSEA was also applied to the control samples (6 hr, 72 hr, & 240 hr) to determine developmental or tissue-specific expression profiles. Following GSEA, we performed leading-edge analysis which focuses on the core gene members that account for the gene set's enrichment signal as not all members of the gene-set will typically participate in a biological process underlying a disease phenotype. This means that the genes that contribute most to a given pathway's enrichment (i.e., core genes) will be located at the top (most upregulated) or bottom (most downregulated) of the ranked gene list (*Subramanian et al., 2005*).

To overcome gene-set redundancy and help in the interpretation of large pathway lists, clustering of GSEA results was then performed using EnrichmentMap (*Merico et al., 2010*) for each experimental group (normal development, FDMI and FDMR) and an overlap similarity coefficient cut-off of 0.5.

## RESULTS

### Ocular biometrics for FDMI and FDMR

We collected refractive state (Rx, D), vitreous chamber depth (VCD, mm) and axial length (AL, mm) measures for all timepoint conditions (FDMI, FDMR, control) as biometric measures were not available for McGlinn and colleagues' original microarray dataset at 6 hr and 72 hr of FDMI. Thus, the earlier biometric measures presented in Fig. 2 for 6 hr and 72 hr induction datasets were included to provide relative indicators of growth and refraction changes expected at that number of hours of occlusion.

In response to occluder wear, AL and VCD increased and refraction became more negative with time post-occlusion under our laboratory conditions. There was a significant main effect for FD induction on refraction [$F(1, 20) = 33.16, p < .001$] and time [$F(1, 20) = 17.99, p < .001$]. A significant interaction was also observed between FD and time [$F(1, 20) = 10.67, p < .001$]. A significant main effect was observed for FD induction on axial length measurements [$F(1, 20) = 15.40, p < .001$] and time [$F(1, 20) = 33.17, p < .001$]. A significant main effect was observed for time for VCD [$F(1, 20) = 4.31, p < .05$] but not for FD induction [$F(1, 20) = 3.61, p = .07$].

For our recovery dataset (10 d control, 10 d FDMI, 6 hr FDMR and 24 hr FDMR), biometric measurements and gene expression profiles were collected from the same chicks. Both refractive status and axial dimensions normalised rapidly following occluder removal

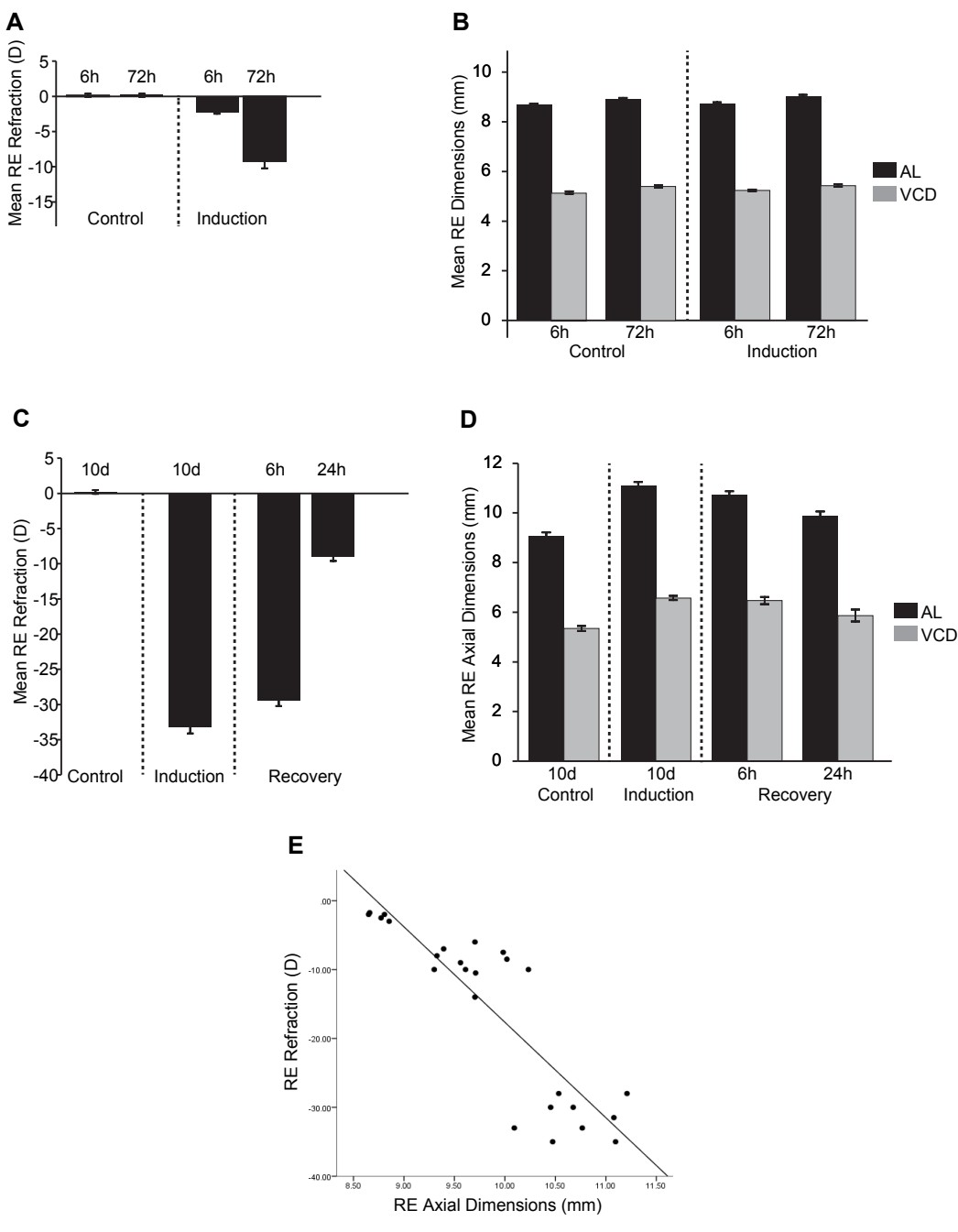

**Figure 2  Ocular Biometrics for FDMI and FDMR.** Mean (±SE) measures of refractive status, axial length (AL) and vitreous chamber depth (VCD). To complement the data reported by *McGlinn et al. (2007)*, (A) refraction and (B) AL & VCD were collected during 6 h and 72 h of normal development and 6 h and 72 h following seven-days induction of myopia. Refraction, AL & VCD measures for 24 h recovery after prolonged (i.e., 10 days) form deprivation is shown in (C) and (D). Both refractive state and axial length changes were highly correlated ($r = .78$) during occluder wear (E). Note: Measures for anterior chamber and lens thickness are included in Fig. S1.

and 24 hr of normal visual experience (Fig. 2). These measures were shown to be highly correlated ($r = .78$; Fig. 2E).

There was a significant main effect on refraction in FD recovery [$F(1, 20) = 1499.27, p < .001$] and time [$F(2, 20) = 134.0, p < .001$]. A significant interaction was also observed between FD and time [$F(2, 20) = 141.46, p < .001$]. Post-hoc tests revealed that refraction was significantly different at all induction time-points ($p < .01$) except between 10 days of occlusion (0 h) and 6 h recovery ($p = .13$). Axial length was significantly different for FD recovery [$F(1, 20) = 199.87, p < .001$] and time [$F(2, 20) = 7.07, p < .01$]. A significant main effect for FD was observed for VCD [$F(1, 20) = 146.49, p < .001$] but no main effect was seen for time [$F(2, 20) = 2.275, p = .13$].

MRI of the FD chick at 72 hr post-recovery shows substantial retinal thinning and choroidal expansion. These observations confirm our previous histological work (*Liang et al., 2004*) where maximum choroidal expansion in the FD eye is approximately 300% greater than the fellow eye

## Gene set enrichment analysis

To understand the underlying biological processes governing myopia development we performed pathways enrichment analysis using the Gene Set Enrichment Analysis (GSEA). We focused on identifying KEGG, Reactome, and BioCarta pathways showing expression shifts that correlated with the time post-FD by occlusion (in hours). We performed three separate analyses to deal with the complexity of the data including sample pooling. The first analysis involved normalised log expression data for short-term FD (6 hr and 72 hr) induction. The second dataset included averaged log expression for when data was collected during the combined long-term induction period (i.e., 6 hr, 72 hr, and 240 hr (10 d) for control and FDMI datasets) and thirdly, analysis was performed on average expression values during recovery post FD occlusion (i.e., 0 hr (post-occlusion), 6 hr, and 24 hr).

### Enrichment analysis for short-term FDM induction

Our initial analysis focused on identifying KEGG, Reactome, and BioCarta pathways showing expression shifts that correlated with short-term FD to compliment previous findings by *McGlinn et al. (2007)*. Analysis of gene expression changes between 6 hr and 72 hr of FD revealed 13 significantly enriched pathways. Most pathways responding to short-term FD involve mitochondrial energy metabolism (Table 1). Interestingly, none of the genes identified by *McGlinn et al. (2007)* were listed as a core gene for these pathways suggesting that highly regulated genes are not always driving treatment-specific biological responses. Notably, genes involved in 'One carbon pool by folate' was significantly upregulated suggesting an association between homocysteine (Hcy) and B-vitamins in short-term FD induction. Genes associated with short-term FD induction have previously been associated with AMD (*Gopinath et al., 2013*).

### Enrichment analysis for long-term FDM induction (FDMI) and recovery (FDMR)

We next performed GSEA to identify KEGG, Reactome, and BioCarta pathways showing expression shifts that correlated with longer-term FD by occlusion (in hours; 6 hr, 72 hr,

**Table 1  Pathways enriched between 6 h and 72 h of FD induction.** Mean Normalised Enrichment Score (NES) and false discovery rate (FDR) for the biological pathways identified by GSEA in FD induction. Most genes responding to FD are involved in mitochondrial energy metabolism. The NES reflects the degree to which a set of genes is over-represented at either the top or bottom of a ranked list of genes while also taking into account differences in pathway size (i.e. gene-set size) and is the primary statistic for examining enrichment results, and for comparing results across pathways.

| Pathway | Core genes | Database | NES | FDR |
|---|---|---|---|---|
| Huntingtons disease | APAF1, ATP5C1, ATP5D, ATP5E, ATP5F1, ATP5H, ATP5O, BDNF, CASP3, CASP8, CLTA, COX4I1, COX5A, COX6A1, COX6C, COX7A2, COX7A2L, COX7C, COX8A, CYCS, DNAH3, DNAI1, DNAL1, DNALI1, GPX1, HDAC2, IFT57, NDUFA1, NDUFA10, NDUFA2, NDUFA4, NDUFA5, NDUFA6, NDUFA7, NDUFA8, NDUFB1, NDUFB10, NDUFB3, NDUFB4, NDUFB5, NDUFB6, NDUFB8, NDUFB9, NDUFC1, NDUFC2, NDUFS1, NDUFS3, NDUFS4, NDUFS6, NDUFS7, NDUFS8, NDUFV3, PLCB1, PLCB4, POLR2D, POLR2F, POLR2J, POLR2L, PPID, SDHA, SLC25A4, SLC25A6, SOD2, TAF4, TBP, TBPL1, TFAM, UQCR10, UQCR11, UQCRFS1, UQCRH, VDAC3 | KEGG | 1.76 | 0.15 |
| Oxidative phosphorylation | ATP5C1, ATP5D, ATP5E, ATP5F1, ATP5H, ATP5I, ATP5J2, ATP5O, ATP6V0D2, ATP6V1G1, COX11, COX15, COX17, COX4I1, COX5A, COX6A1, COX6C, COX7A2, COX7A2L, COX7C, COX8A, NDUFA1, NDUFA10, NDUFA2, NDUFA4, NDUFA5, NDUFA6, NDUFA7, NDUFA8, NDUFB1, NDUFB10, NDUFB3, NDUFB4, NDUFB5, NDUFB6, NDUFB8, NDUFB9, NDUFC1, NDUFC2, NDUFS1, NDUFS3, NDUFS4, NDUFS6, NDUFS7, NDUFS8, NDUFV3, PPA1, PPA2, SDHA, UQCR10, UQCR11, UQCRFS1, UQCRH | KEGG | 1.81 | 0.16 |
| Mitochondrial protein import | BCS1L, CHCHD4, COX17, DNAJC19, GRPEL1, HSCB, HSPD1, PAM16, PMPCA, PMPCB, SAMM50, SLC25A12, SLC25A4, SLC25A6, TIMM13, TIMM17A, TIMM22, TIMM44, TIMM9, TOMM22, TOMM5, TOMM7 | Reactome | 1.80 | 0.16 |
| One carbon pool by folate | ATIC, DHFR, GART, MTFMT, MTHFD1, MTHFD1L, MTHFS, TYMS | KEGG | 1.74 | 0.17 |
| Cholesterol biosynthesis | DHCR7, FDFT1, GGPS1, HMGCR, HMGCS1, IDI1, MSMO1, NSDHL, SQLE | Reactome | 1.77 | 0.17 |
| Antigen processing cross presentation | CD36, CTSS, NCF4, PSMA1, PSMA2, PSMA3, PSMA5, PSMA6, PSMA7, PSMB1, PSMB2, PSMB3, PSMC1, PSMC2, PSMC3, PSMC5, PSMD1, PSMD10, PSMD3, PSMD5, RPS27A, SEC61B, SEC61G, TAP1, UBA52 | Reactome | 1.74 | 0.18 |
| Alzheimers disease | NDUFB6, CYCS, NDUFB3, NDUFA8, UQCR11, NDUFC2, ATP5F1, ATP5E, NDUFB5, UQCR10, COX4I1, COX6A1, COX7C, NDUFB10, NDUFB1, CASP8, NDUFV3, NDUFA6, ATP5H, COX7A2, RYR3, NDUFA2, NDUFC1, NDUFS1, NDUFA1, SDHA, NDUFB8, NDUFA4, NDUFA5, APAF1, UQCRH, ATP5O, NDUFA7, NDUFS6, NDUFS4, ATP5D, ATP5C1, NDUFB9, CACNA1D, NDUFS8, NDUFS3, NDUFS7, CASP3, COX8A, COX6C, COX5A, NDUFA10, ATP2A1, COX7A2L, TNFRSF1A, UQCRFS1, PSEN1, NDUFB4, IL1B, PLCB1, NCSTN, PLCB4 | KEGG | 1.74 | 0.19 |
| Parkinsons disease | APAF1, ATP5C1, ATP5D, ATP5E, ATP5F1, ATP5H, ATP5O, CASP3, COX4I1, COX5A, COX6A1, COX6C, COX7A2, COX7A2L, COX7C, COX8A, CYCS, GPR37, NDUFA1, NDUFA10, NDUFA2, NDUFA4, NDUFA5, NDUFA6, NDUFA7, NDUFA8, NDUFB1, NDUFB10, NDUFB3, NDUFB5, NDUFB6, NDUFB8, NDUFB9, NDUFC1, NDUFC2, NDUFS1, NDUFS3, NDUFS4, NDUFS6, NDUFS7, NDUFS8, NDUFV3, PARK7, PPID, SDHA, SLC25A4, SNCAIP, UBE2L3, UQCR10, UQCR11, UQCRH, VDAC3 | KEGG | 1.77 | 0.19 |
| TCA cycle and respiratory electron transport | ATP5C1, ATP5D, ATP5E, ATP5F1, ATP5H, ATP5I, ATP5J2, ATP5O, COX4I1, COX5A, COX6A1, COX6C, COX7A2L, COX7C, COX8A, CYCS, D2HGDH, DLD, IDH3A, LDHB, NDUFA1, NDUFA10, NDUFA2, NDUFA4, NDUFA5, NDUFA6, NDUFA7, NDUFA8, NDUFB1, NDUFB10, NDUFB3, NDUFB5, NDUFB6, NDUFB8, NDUFB9, NDUFC1, NDUFC2, NDUFS1, NDUFS3, NDUFS4, NDUFS6, NDUFS7, NDUFS8, NDUFV3, NNT, SDHA, SUCLG1, SUCLG2, UQCR11, UQCRH | Reactome | 1.72 | 0.19 |

**Table 1** (*continued*)

| Pathway | Core genes | Database | NES | FDR |
|---|---|---|---|---|
| Respiratory electron transport | COX4I1, COX5A, COX6A1, COX6C, COX7A2L, COX7C, COX8A, CYCS, NDUFA1, NDUFA10, NDUFA12, NDUFA2, NDUFA4, NDUFA5, NDUFA6, NDUFA7, NDUFA8, NDUFB1, NDUFB10, NDUFB3, NDUFB4, NDUFB5, NDUFB6, NDUFB8, NDUFB9, NDUFC1, NDUFC2, NDUFS1, NDUFS3, NDUFS4, NDUFS6, NDUFS7, NDUFS8, NDUFV3, SDHA, UQCR11, UQCRFS1, UQCRH | Reactome | 1.81 | 0.21 |
| Respiratory electron transport ATP synthesis by chemiosmotic coupling and heat production by uncoupling proteins | ATP5C1, ATP5D, ATP5E, ATP5F1, ATP5H, ATP5I, ATP5J2, ATP5O, COX4I1, COX5A, COX6A1, COX6C, COX7A2L, COX7C, COX8A, CYCS, NDUFA1, NDUFA10, NDUFA12, NDUFA2, NDUFA4, NDUFA5, NDUFA6, NDUFA7, NDUFA8, NDUFB1, NDUFB10, NDUFB3, NDUFB4, NDUFB5, NDUFB6, NDUFB8, NDUFB9, NDUFC1, NDUFC2, NDUFS1, NDUFS3, NDUFS4, NDUFS6, NDUFS7, NDUFS8, NDUFV3, SDHA, UCP3, UQCR11, UQCRFS1, UQCRH | Reactome | 1.83 | 0.23 |
| Translation | EEF1B2, EIF2B1, EIF2B2, EIF2S1, EIF2S2, EIF2S3, EIF3D, EIF3H, EIF3I, EIF3J, EIF4EBP1, EIF4H, EIF5B, RPL10A, RPL11, RPL13, RPL14, RPL18A, RPL19, RPL21, RPL22, RPL23A, RPL24, RPL26L1, RPL27, RPL27A, RPL29, RPL30, RPL32, RPL35, RPL35A, RPL36, RPL36A, RPL37, RPL37A, RPL38, RPL39, RPL5, RPL6, RPL7, RPL8, RPLP1, RPLP2, RPN1, RPS10, RPS11, RPS14, RPS15, RPS15A, RPS16, RPS2, RPS20, RPS23, RPS24, RPS25, RPS26, RPS27A, RPS28, RPS29, RPS3, RPS3A, RPS4X, RPS6, RPS7, RPS8, RPSA, SEC61B, SEC61G, SPCS1, SPCS2, SPCS3, SRP19, SRP72, SSR3, UBA52 | Reactome | 1.69 | 0.24 |
| ER phagosome pathway | PSMA1, PSMA2, PSMA3, PSMA5, PSMA6, PSMA7, PSMB1, PSMB2, PSMB3, PSMC1, PSMC2, PSMC3, PSMC5, PSMD1, PSMD10, PSMD3, PSMD5, RPS27A, SEC61B, SEC61G, TAP1, UBA52 | Reactome | 1.67 | 0.25 |

and 240 hr (10 d) for control and FDMI datasets; and 0 hr (post-occlusion), 6 hr, and 24 hr for the FDMR dataset. This analysis identified 61 significantly enriched pathways during normal development, 130 during FDMI, and one during FDMR. We then used the Enrichment Map (*Merico et al., 2010*) to cluster highly similar pathways into networks. Many pathways were implicated during both normal (control) development and FDMI suggesting that most of these pathways are likely to relate more to rate of growth rather than to the form-deprivation paradigm *per se.* That is, given that the same pathways were identified irrespective of the experimental manipulation it is likely that similar mechanisms associated with growth are likely to operate under any experimental conditions. The direction of regulation i.e., up or down, would be expected to be associated with rates of growth and with the age of the chicks and would be predicted by the RIDE model. Summary statistics for the resulting clusters are presented in Table 2, and the clusters are visualized in Figs. 3 and 4.

FDMI was characterised by 19 clusters of pathways (Fig. 4). Six of these clusters were also altered during normal ocular development ('cell cycle, mitotic', 'cytochrome p450', 'neurotransmission', 'phospholipid metabolism', 'signal transduction, GPCR' and 'vesicle-mediated transport') but demonstrate greater signal strength during FDMI compared to normal development (Table 2). The direction of expression change for these clusters is consistent across both normal development and FDMI.

**Table 2  Summary statistics for clusters of pathways enriched during FDMI and normal development.** Mean Normalised Enrichment Score (NES), false discovery rate (FDR) and signal strength statistic (Signal) for the biological pathways implicated by GSEA in control and FDM. Normal eye development implicated 10 cluster of pathways showing average signal strength while form-deprivation induction implicated 18 clusters of pathways. Pathways shown here only include clustered pathways as represented in Figs. 3 and 4 and do not include pathways that were unclustered. Further detail on the unclustered pathways can be found in File 1. The NES reflects the degree to which a set of genes is overrepresented at either the top or bottom of a ranked list of genes while also taking into account differences in pathway size (i.e., geneset size). NES is the primary statistic for examining enrichment results, and for comparing results across pathways. The percentage signal strength statistic reflects the proportion of the core set of genes that contribute most to a given pathway's enrichment by accounting for particular genes position in the ranked list. A high signal strength indicates that the genes within a pathway are located close to the top (positive NES) or bottom (negative NES) of the ranked gene list. If the core genes are spread throughout the ranked list, then the signal strength decreases towards zero (Mootha et al., 2003; Subramanian et al., 2005).

| Cluster | Control | | | | FDMI | | | |
|---|---|---|---|---|---|---|---|---|
| | Pathways in cluster | NES | FDR q-value | Signal | Pathways in cluster | NES | FDR q-value | Signal |
| Cell cycle, mitotic | 5 | −2.233 | 0.019 | 31% | 3 | −2.253 | 0.009 | 98% |
| Cell maintenance & survival | – | – | – | – | 28 | −2.180 | 0.014 | 117% |
| Clatherin-mediated endocytosis (CME) | – | – | – | – | 2 | −2.014 | 0.032 | 81% |
| Complement and coagulation cascades (CCC) | – | – | – | – | 2 | 2.708 | 0.003 | 140% |
| Cytochrome p450 | 2 | 2.247 | 0.025 | 47% | 5 | 2.629 | 0.002 | 133% |
| Cytokine pathways | 2 | 2.717 | 0.002 | 40% | – | – | – | – |
| Neuron structure/growth | – | – | – | – | 5 | −2.321 | 0.007 | 80% |
| Fatty acid (FA) metabolism | – | – | – | – | 2 | −1.886 | 0.032 | 84% |
| Glucosaminoglycan (GAG) metabolism | 4 | 2.121 | 0.029 | 40% | – | – | – | – |
| Ion channel transport | – | – | – | – | 2 | −2.331 | 0.007 | 99% |
| Mitochondrial energy metabolism | – | – | – | – | 7 | 2.396 | 0.005 | 47% |
| Neurotransmission | 5 | −2.522 | 0.011 | 37% | 8 | −2.423 | 0.007 | 103% |
| Peroxisome | 2 | −2.221 | 0.017 | 40% | – | – | – | – |
| Phospholipid metabolism | 2 | −2.134 | 0.026 | 28% | 3 | −1.939 | 0.027 | 57% |
| Signal transduction, growth factors (GF) | – | – | – | – | 4 | −1.984 | 0.021 | 66% |
| Signal transduction, g-protein coupled receptors (GPCR) | 7 | 2.572 | 0.019 | 52% | 7 | 2.683 | 0.010 | 122% |
| Signal transduction, mitogen-activated protein kinases (MAPK) | – | – | – | – | 4 | −1.931 | 0.028 | 67% |
| Signal transduction, nerve growth factor (NGF) | – | – | – | – | 5 | −2.195 | 0.014 | 84% |
| Transcription | – | – | – | – | 10 | −2.218 | 0.013 | 120% |
| Translation | – | – | – | – | 8 | 2.731 | 0.001 | 46% |
| Ubiquitin-mediated proteolysis | 3 | −2.491 | 0.006 | 39% | – | – | – | – |
| Vesicle-mediated transport | 3 | −2.152 | 0.025 | 40% | 3 | −2.097 | 0.013 | 103% |

## Leading edge analysis

We next examined the leading-edge subsets (i.e., core genes) within the clusters identified during FDMI, for potential genes that may play a role in the broader ultrastructural evidence of oxidative and hyperosmotic stress previously described during FDM (Crewther et al., 2006; Liang et al., 1995; Liang et al., 2004) and in brain (Brocker, Thompson & Vasiliou, 2012; Morland, Pettersen & Hassel, 2016; Veltmann et al., 2016). These clusters included mitochondrial metabolism gene sets previously identified by Riddell et al. (2016) in similar aged chicks with refractive errors induced by optical defocus.

**Figure 3  Enrichment map for highly clustered pathways in normal eye development.** Gene set enrichment analysis revealed 61 biological pathways that can be functionally grouped into 10 clusters using a coefficient of similarity altered during the 10 days of normal eye development in retina/RPE/choroid. *Note:* Each node represents a biological pathway from File S1. The colour of each node emphasises the direction of expression and normalised enrichment score (NES). Node size is relative to the number of genes in the pathway. Thickness of the connections (green) between each node reflects the degree of similarity between each gene set. Twenty-six pathways did not meet the clustering similarity coefficient of 0.5 and hence are not shown here. *Note cluster names: GAG, glycosaminoglycan; GPCR, g-protein coupled receptors.*

Gene sets involved in maintaining mitochondrial metabolism (oxidative phosphorylation, TCA cycle and respiratory electron transport/ATP synthesis) or diseases involving mitochondrial dysfunction (i.e., Huntington's disease, Parkinson's and Alzheimer's Diseases) were significantly upregulated during short-term FDM (Table 1) and long-term FDMI (NES = 2.40, $p = .005$; Table 2), and displayed non-significant down-regulation during the 24 hr period of FDMR (Fig. 4). Figure 5 illustrates the relative changes in these particular pathways at different time-points. By comparison, there were no significant changes in expression of these mitochondrial metabolism pathways during normal ocular development (Table 2). This shift in metabolic regulation during induction of FDM and recovery is reminiscent of the bidirectional regulation of metabolic genes described for signed optical defocus by (*Riddell et al., 2016*).

Examination of the leading-edge subsets (i.e., core genes) of the mitochondrial metabolism pathways revealed 72 high scoring core genes that contributed to its significance during FDMI (File S1B). Fourteen of these core genes are present in all mitochondrial energy metabolism gene sets and form part of mitochondrial complex I (NDUFA1, NDUFA4, NDUFA5, NDUFA6, NDUFB1, NDUFB3, NDUFB6, NDUFB9, NDUFS2, NDUFS3, NDUFS7), complex III (UQCR11, UQCRH) and complex IV (COX7C). Both complex I and III are primary producers of reactive oxygen species (ROS) in neural tissues resulting in oxidative stress when cells are unable to stabilise the ROS (*Murphy, 2009*). Furthermore, an

**Figure 4** **Enrichment map for highly clustered pathways in form deprivation induction and recovery.** Axial elongation during 10 days of form-deprivation compared to normal unoccluded controls resulted in 130 altered pathways in retina/RPE choroid (inner node) while 24 h recovery (outer annulus) identified only one statistically significant pathway i.e., bile acid & bile salt metabolism. Pathways not statistically enriched during FDMR are shown for comparison purposes. Notably, expression profiles of FDMI and FDMR are consistent despite the fact that only the FDMR data includes choroidal tissue. Pathways highly expressed during induction (red inner node) were often suppressed during normal vision and recovery (blue outer annulus) and vice versa. *Note:* Each node represents a biological pathway from File S1. The colour of each node emphasises the direction of expression and normalised enrichment score (NES). Node size is relative to the number of genes in the pathway. Thickness of the connections (green) between each node reflects the degree of similarity between each gene set. There were 22 unclustered pathways in FDMI that did not meet the clustering similarity coefficient of 0.5. *Note cluster names: CCC, complement and coagulation cascades; CME, clatherin-mediated endocytosis; FA, Fatty acid; GF, growth factors; GPCR, g-protein coupled receptors; MAPK, mitogen-activated protein kinases; NGF, nerve growth factor.*

additional 11 core genes were present in six out of seven gene sets with a majority coding for mitochondrial complex V subunits that catalyse the conversion of adenosine diphosphate (ADP) and inorganic phosphate (Pi) to ATP. Taken together, these data strongly implicate mitochondrial metabolism in the physiology of myopia.

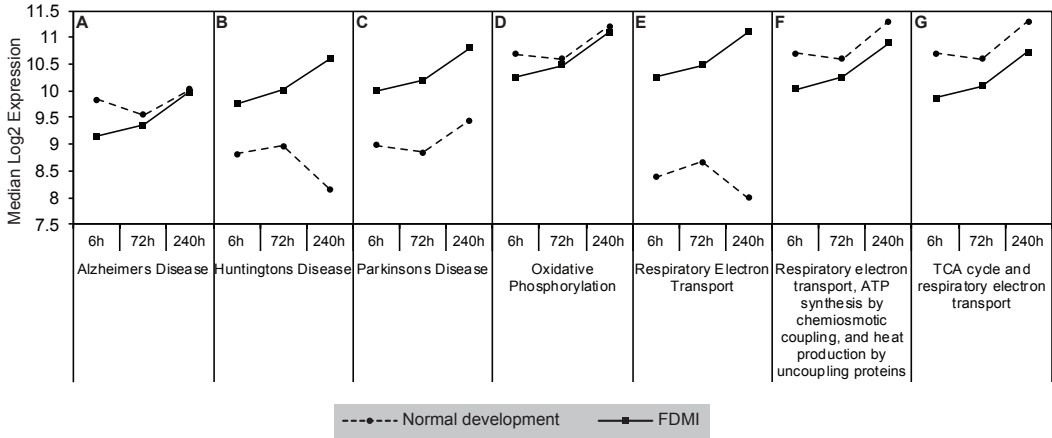

**Figure 5  Median expression of pathways involved in mitochondrial metabolism.** Graphs of the seven mitochondrial metabolism pathways with significant expression shifts across 240 h of occluder wear relative to unoccluded controls (A) Alzheimer's disease, (B) Huntington's disease, (C) Parkinson's disease, (D) Oxidative phosphorylation, (E) Respiratory electron transport, (F) Respiratory electron transport/ATP synthesis by chemiosmotic coupling and heat production by uncoupling proteins, (G) TCA cycle and respiratory electron transport.

A similar analysis of leading edge subsets relating to neurotransmission demonstrated significant negative correlations with duration of occluder wear (NES = −2.42, p = 0.007; Table 2 and Fig. 6). While these pathways showed transcriptional activation during FDMR, they did not reach the FDR cut-off of 0.05 during the short recovery period.

Five neurotransmission-related gene sets also reported significant expression shifts during normal ocular development (Table 2; Fig. 6). The significance in enrichment of these pathways during normal eye development was the result of 124 core genes with 10 genes common in four out of five pathways (ARHGEF9, GABRA1, GABRA2, GABRA6, GABRB2, GABRB3, GABRG2, GABRG3, GABRR1, GABRR2; File S1A). In long-term FDMI, the signal strength rose to 103% (Table 2) for neurotransmission for which there were 139 core genes (seven genes common in seven out of eight pathways). The seven genes in FDMI were serine/threonine protein kinases (BRAF, CAMK2B), ionotropic NMDA glutamate receptors (GRIN1, GRIN2A), mitogen-activated protein kinase (MAPK1) and ribosomal protein S6 kinase (RPS6KA2, RPS6KA6). In conjunction with a decrease in neurotransmission, analysis revealed suppression in clatherin-mediated endocytosis during FDMI (NES = −2.014, p = 0.032; Table 2) suggesting that prolonged FDMI results in a suppression in synaptic transmission.

Notably, a cluster of two ion-transport related pathways showed commonalities in genes which in turn were negatively correlated with long-term occluder wear (Fig. 7). Indeed, the 'ligand-gated ion channel' pathway was clustered with the neurotransmission-related pathways during normal development. The leading-edge analysis identified 15 genes that contribute to the enrichment of these pathways (ARHGEF9, GABRA1, GABRA2, GABRA5, GABRA6, GABRB2, GABRB3, GABRG2, GABRG3, GABRR1, GABRR2, GLRA1, GLRA2,

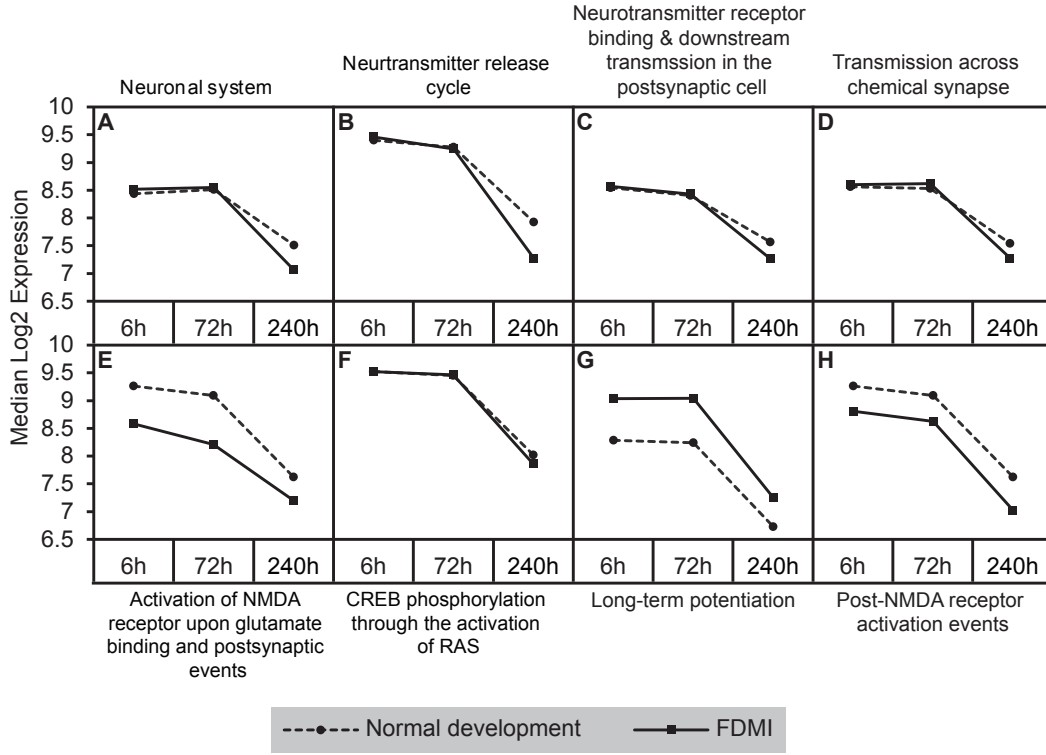

**Figure 6 Median expression of pathways involved in neurotransmission during normal ocular development and in FDMI.** Graphs of the neurotransmission-related pathways with significant expression shifts during normal ocular development (dotted lines) and FDMI (solid lines) are shown. (A–D) Four pathways were significant in both normal development and FDMI. The leading-edge subsets for these pathways identified 115 common core genes shared within these pathways during normal development and during FDMI and 27 other core genes specific to normal development and nine specific to FDMI (File S1). (E–H) Graphs indicate FDMI induced down regulation of expression shift in four additional neurotransmission-related pathways with significant expression shifts during FDMI (solid lines) only. These pathways were not significant during normal ocular development but data are shown for comparison purposes (dotted lines). (A) Neuronal system (B) Neurotransmitter release cycle (C) Neurotransmitter receptor binding & downstream transmission in the postsynaptic cell (D) Transmission across chemical synapse (E) Activation of NMDA receptor upon glutamate binding and postsynaptic events (F) CREB phosphorylation through the activation of RAS (G) Long-term potentiation (H) Post-NMDA receptor activation events.

GLRA3, GLRB) implicating chloride currents in the development of form-deprivation myopia (*Zhang et al., 2011*). This is of particular interest, as one of the aims of this study was to determine if pathways related to the mitochondrial abnormalities and ion redistribution patterns observed in our previous ultrastructural and elemental microanalysis work (*Crewther et al., 2006*; *Liang et al., 2004*; *Liang et al., 1996*) were also identifiable using the GSEA approach to microarray data.

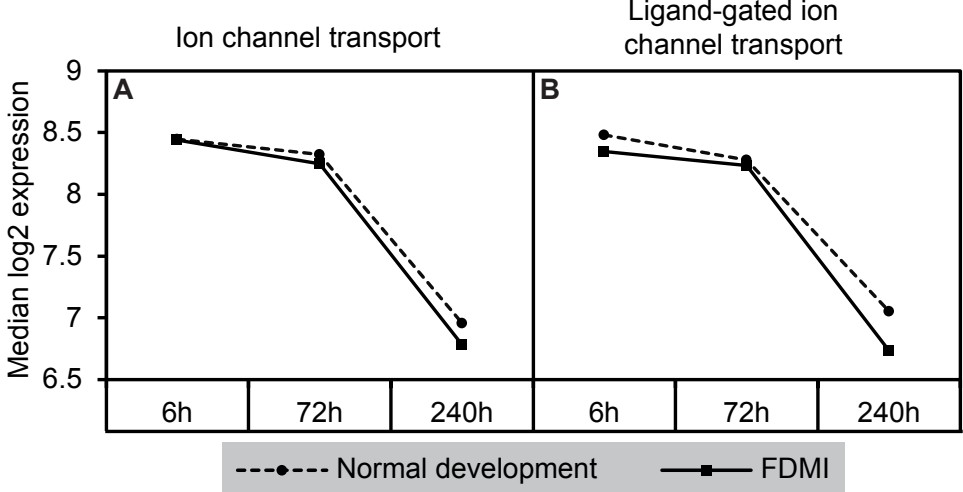

**Figure 7** **Median expression of pathways involved in ion transport during normal ocular development and in FDMI.** Expression of the (A) Ion channel transport and (B) Ligand-gated ion channel transport pathways with significant expression shifts during FDMI (solid lines) compared to normal development (dotted lines) are shown. The '*Ligand-gated ion channel transport pathway*' pathway was also significantly altered during normal development and was clustered with the neurotransmission pathway.

## Novel pathways associated with long-term FDMI as identified by GSEA

### Complement and coagulation cascades

The strongest median signal of 140% and NES of 2.71 during FDMI (Table 2) was found for two pathways involved in the complement and coagulation cascades (Fig. 8). The Reactome '*Complement and coagulation cascade*' pathway was one of the two pathways within this cluster also significantly upregulated in controls (NES = 2.63, $p = .001$). The core genes contributing to the upregulation of this cluster of gene sets include 49 genes, with only 17 of these genes recurrent throughout this cluster. These include alpha-2-macroglobulin (A2M), coagulation factors (F2, F3, F7, F8, F9, F10, F11, F13A1, F13B), fibrinogen (FGA, FGG), kininogen (KNG1), serpins (SERPINC1, SERPING1), thrombomodulin (THBD) and von Willebrand factor (VWF).

### Cytochrome p450

This cluster of gene sets produced the second highest median signal (133%) during FDMI with an NES of 2.63 (Table 2; Fig. 9). The core genes underlying the upregulation of this cluster of pathways include the monooxygenases CYP1A2, CYP2C18, CYP3A4, and CYP3A7 that are found in the endoplasmic reticulum (*Park et al., 2014*) and present in all five pathways within this cluster (File S1B). The role of cytochrome p450 is in xenobiotic metabolism and subsequent synthesis of cholesterol, steroids and other lipids (*Nebert & Russell, 2002*). Expression of CYP1A2 is induced by the aryl hydrocarbon receptor (AHR) and HIF 1 beta (*Nebert et al., 2000*; *Stejskalova et al., 2011*). Interestingly, CYP1A1 has been reported to be involved in retinoic acid (RA) biosynthesis where manipulation of the AHR gene results in reduced retinoic acid metabolism (*Andreola et al., 1997*). This is

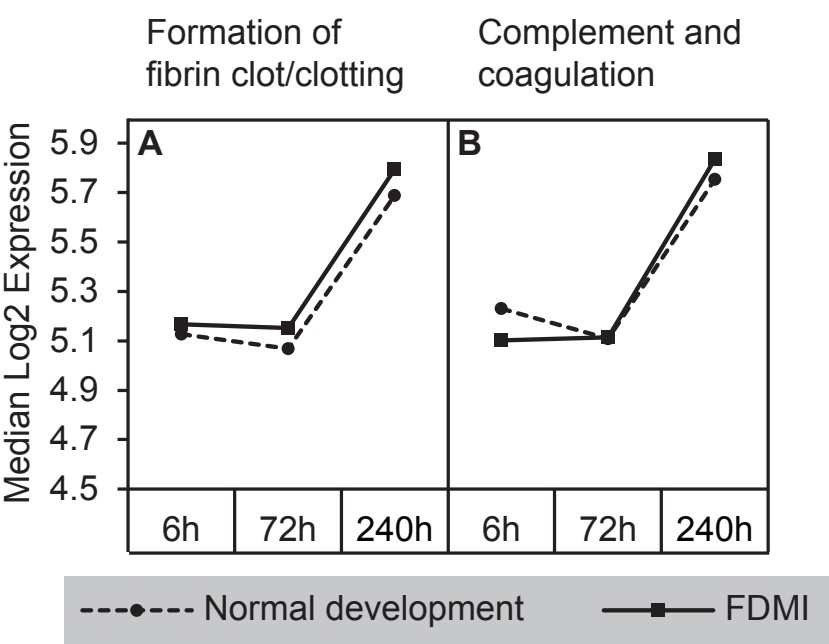

**Figure 8 Median expression of pathways involved in the complement and coagulation cascade.** Graphs indicate greater expression shift in the complement & coagulation cascade between 72 h and 240 h of occluder wear for both (A) formation of fibrin clot/clotting cascade and (B) complement and coagulation cascades. Note that the '*complement and coagulation cascades*' pathway was also significantly altered during normal development.

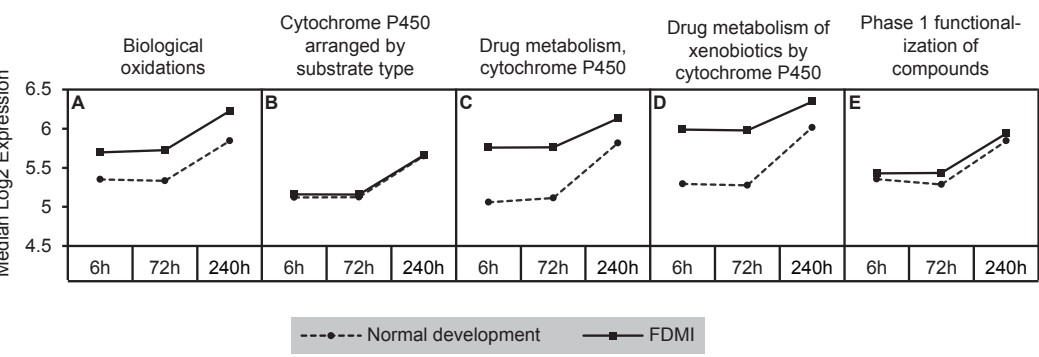

**Figure 9 Median expression of pathways involved in cytochrome p450 metabolism.** Graph indicates enhanced expression in cytochrome p450 related pathways in FDMI compared to normal development. (A) Biological oxidations (B) Cytochrome P450 arranged by substrate type (C) Drug metabolism, cytochrome P450 (D) Drug metabolism of xenobiotics by cytochrome P450 (E) Phase 1 functionalization of compounds.

an interesting finding as CYP7A1 and CYP8B1 have also been identified as core genes in bile acid metabolism during FDMR, and are inhibited by RA (*Yang et al., 2014*). CYP3A4 and CYP3A7 but not CYP2C18 have been shown to be suppressed in the presence of inflammatory cytokines (*Aitken & Morgan, 2007*). Therefore, dysregulation of cytochrome

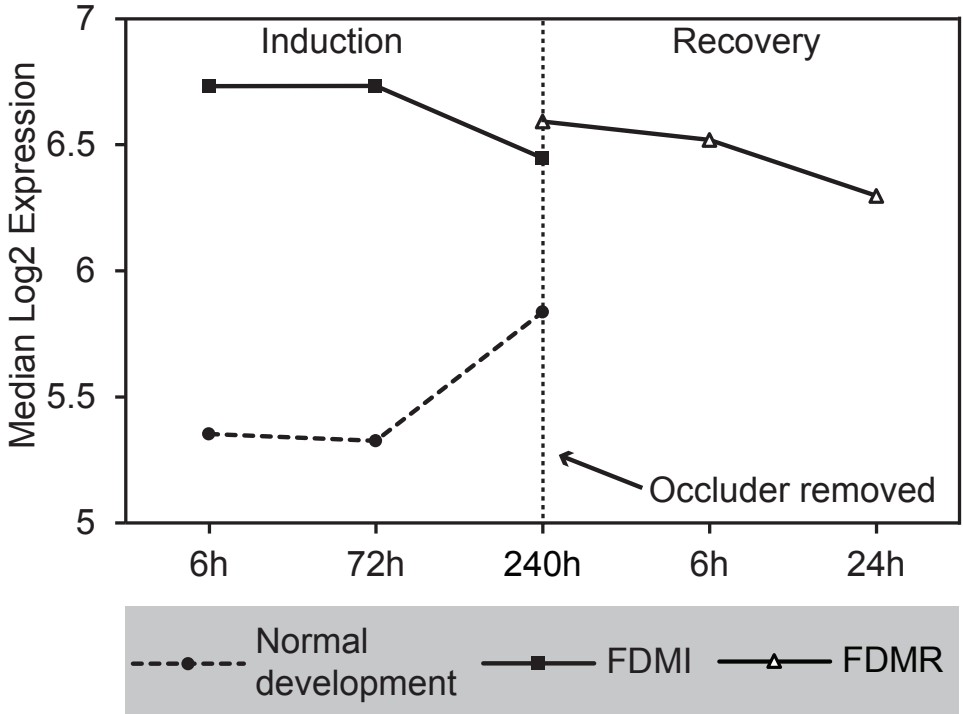

**Figure 10  Median expression of core genes in the bile acid and bile salt metabolism pathway during FDM.** Graphs shows median change of the core genes during normal ocular development, FDMI, and FDMR. This pathway was found to be significant for FDMI (left, solid line) and FDMR (right, solid line) but not in normal development. Note: Median expression value was calculated based on core genes identified in each experimental group. This pathway was not significantly altered during normal ocular development but data are shown for comparison purposes (dotted line).

p450 is likely to have implications for presence of oxidative stress, endoplasmic reticulum stress and, if persistent, to be a serious risk factor for myopia and more severe ophthalmic disease.

## GSEA of recovery from FDM

GSEA to identify KEGG, Reactome, and BioCarta pathways showing expression shifts that correlated with recovery from FD by occlusion (in hours; 6 hr and 240 hr) was also performed. The only gene set significantly altered during the first 24 h of refractive recovery was the bile acid and bile salt metabolism pathway. Bile acid synthesis is reliant on the interaction of peroxisome degradation and mitochondrial metabolism with the end products of cholesterol utilization being the bile acids (*Lefebvre et al., 2009*). Indeed, the synthesis of the bile acids is the major pathway of cholesterol catabolism in mammals. The 14 core genes involved include SLC10A1, ACOX2, AKR1D1, FABP6, CH25H, ABCC3, HSD17B4, ALB, ABCB11, CYP7B1, HSD3B7, SLCO1A2, AMACR, and CYP46A1 (Fig. 10). SLC10A1 (sodium taurocholate cotransporting polypeptide), SLCO1A2 (sodium-independent Organic anion transporter), and cytochrome P450 genes, CYP7B1 and CYP46A1 are known to be involved with cholesterol pathways in brain. Other

genes involved include protein members of the superfamily of ATP-binding cassette (ABC) transporters ABCC3, bile salt export pump ABCB11, and peroxisome genes acyl-coenzyme A oxidase 2 (ACOX2), and hydroxysteroid (17-beta) dehydrogenase 4 (HSD17B4).

### Gene validation

The microarray results described here are consistent with our previously published work using RNA-seq (*Riddell et al., 2016*). Hence, we chose not to validate core genes from each pathway by qPCR or other molecular technique as there are >130 pathways identified in this study. Furthermore, validation of the core genes using qPCR is often questionable as it is reportedly subject to within-lab and technical differences (microarray vs qPCR) (*Nygaard & Hovig, 2009*). Microarrays have been shown to exhibit good sensitivity and specificity in detecting gene expression changes (*Dago et al., 2014*) and perform comparatively to RNA-seq, as indicated above in our lab particularly (*Riddell et al., 2016*). Additionally, single-gene analysis will not confirm significance of biological pathways identified in GSEA. For example, in the FDMI dataset, EIF4EBP1 was highly ranked at the top of the gene list for GSEA (i.e., highly upregulated; see Table S3). This gene was only identified as a core gene in the 'Translation' pathway. A gene that was highly ranked at the bottom of the list (i.e., highly down regulated) was GDAP1. Interestingly, this gene was not listed as a core gene for any of the significant pathways identified by GSEA suggesting that highly regulated genes are not always responsible for driving treatment-specific biological responses.

## DISCUSSION

Application of the GSEA technique to the existing FDMI microarray data sets and to our new FDMR dataset demonstrate generalized statistical differences in gene pathways. Mitochondrial energy metabolism was the predominant pathway upregulated in both short-term and long-term FDMI. Long-term FDMI resulted in further dysregulation in several pathways including suppression of neurotransmission, neuron structure/growth and subsequent ion transport compared to normal development. Complement pathways were also upregulated significantly over the time of form-deprivation. Furthermore, the only pathway significantly altered during the first 24 h of refractive recovery was the bile acid and bile salt metabolism pathway. This pathway is reliant on the interaction of peroxisome degradation, fatty acid and mitochondrial metabolism (*Poirier et al., 2006*) and is consistent with the downregulation of fatty acid and PPAR pathways observed in chicks after one day of positive-lens defocus (*Riddell et al., 2016*).

The identification of significantly upregulated mitochondrial energy metabolism pathways is an important finding. While there was a tendency towards upregulation of the mitochondrial energy metabolism pathway during normal development, these pathways were consistently upregulated immediately after the induction of occlusion and then gradually increased again in expression during later FDMI times (Figs. 4 and 5) as refractive compensation and growth rates normalized. The upregulation in mitochondrial energy pathways over the 10 days of FDMI was not unexpected given previous ultrastructural evidence of abnormal photoreceptor elongation and mitochondrial loss of integrity (*Beresford, Crewther & Crewther, 1998*; *Liang et al., 1995*; *Liang et al., 2004*), and expression

studies providing evidence for altered energy metabolism (*Riddell et al., 2017*; *Riddell et al., 2016*) and oxidative stress (*Francisco, Salvador & Amparo, 2015*; *Riddell & Crewther, 2017b*) in myopia.

The leading-edge genes identified as responsible for the change in these pathways primarily code for the mitochondrial complexes I and III that are primary producers of ROS in the brain and are associated with inability to stabilise ROS. Such instability is known to result in oxidative stress (*Francisco, Salvador & Amparo, 2015*; *Murphy, 2009*) which is a likely explanation for the cellular and mitochondrial damage in the retina previously demonstrated ultrastructurally in the FDM model (*Liang et al., 2004*; *Liang et al., 1996*). Defects in mitochondrial complexes I and III as a result of increased superoxide and other reactive oxygen species production (*Adam-Vizi, 2005*) have also been shown to lead to neurodegeneration and subsequent vision loss (*Yu et al., 2012*). Indeed, this finding is consistent with recently published evidence of increased expression of TCA cycle and mitochondrial metabolism genes following negative-lens wear in chicks (*Riddell et al., 2016*) and disruptions to TCA cycle metabolite abundance following FDMI in guinea pigs (*Yang et al., 2017*) further cementing the importance of the mitochondrial respiratory electron transport chain machinery in myopia development.

GSEA also demonstrated a greater number of suppressed transcription and signal transduction pathways during FDMI (Fig. 4) compared to normal development (Fig. 3). The presence of cell maintenance and survival pathways identified during FDMI suggests coordinated interactions between transcription factors, cell cycle components, and signalling molecules (*Rue & Martinez Arias, 2015*), as would be expected to change in a system responding to external stimuli. The majority of genes underlying the transcriptional events in FDMI were proteasome subunit genes (File S1B). The fact that these genes were suppressed during occluder wear and reduced blood flow further highlights a system under severe physiological and oxidative stress as activation of the proteasome promotes cell survival against ROS-mediated oxidative stress (*Choi et al., 2016*) as predicted by the RIDE model.

Recent evidence has also suggested a role for ROS in signal transduction by mediating a variety of cellular processes (*Sena & Chandel, 2012*) including the regulation of neurotransmission (*Wilson & Gonzalez-Billault, 2015*), NMDA receptor-mediated plasticity (reviewed in *Borquez et al., 2016*), modification of ion transport mechanisms (Cl channels and cell swelling (*Liu et al., 2009*)), endoplasmic reticulum (ER) stress and apoptosis by inhibiting WNT activation (*Shen et al., 2014*). Indeed, when mitochondrial metabolic pathways were upregulated during FDMI, NMDA-mediated signalling and ion transport pathways appeared down-regulated, implicating altered glutamate and glycine signalling, water transport and chloride distribution during FDMI. This NMDA signalling has previously been implicated in ocular growth control in experimental myopia (*Fischer, Seltner & Stell, 1997*; *Fischer, Seltner & Stell, 1998*) and together are predicted by the RIDE model.

Accumulating evidence suggests co-influencing roles between oxidative stress and one-carbon metabolism (1-C). In a mouse model of Parkinson's disease, high Hcy levels inhibited mitochondrial complex 1 activity subsequently leading to an increase

in oxidative stress and loss of dopaminergic neurons in the substantia nigra (*Paul et al., 2018*). These findings may have implications in the current understanding of the role of dopamine in myopia (*Zhou et al., 2017*). Folate deficiency has been associated with many ocular abnormalities including ectopic lentis, secondary glaucoma, optic atrophy, retinal detachment, cataracts, retinal vascular occlusive disease (*Ramakrishnan et al., 2006*) and AMD (*Gopinath et al., 2013*). In astronauts, high Hyc and low folate levels are associated with ophthalmic changes after space flight (*Zwart et al., 2012*). The microgravity-fluid shifts experienced by astronauts during space flights result in greater choroidal expansion, refractive changes and abrupt increases in IOP (*Lee et al., 2016*) similar to that observed in the chick model of FDM (*Liang et al., 2004*) further supporting the RIDE model (*Crewther, 2000*; *Crewther et al., 2006*). It is not yet known whether treatment with folic acid will reverse ocular abnormalities where Hcy is elevated however Hcy levels may be an early indicator of myopia (*Yap & Naughten, 1998*).

The identification of the complement and coagulation cascade as having highest signal strength in comparison to other clusters during prolonged occlusion and induction of FDM implicates previously described physiological stress mechanisms associated with constriction of the choroid and reduced blood flow (*Shih, Fitzgerald & Reiner, 1993b*). This is an important result as the relationship between complement factors and myopia has only been reported once in humans (*Long et al., 2013*) and in the cells from posterior sclera of experimentally-induced myopia in guinea pigs (*Gao, Long & Yang, 2015*). More recently, a meta-analysis has suggested a role for the complement system in experimental myopia (*Riddell & Crewther, 2017b*). Notably one of the core genes identified in this pathway was serpin peptidase G (C1 inhibitor; SERPING1), which is reported to function to maintain blood vessel integrity by binding to F12a (not identified in this study) and inhibiting Bradykinin (*Davis et al., 1986*), a protein that promotes inflammation by increasing the permeability of blood vessel walls (*Greenwood, 1991*) and calcium-dependent release of glutamate from astrocytes (*Parpura et al., 1994*). Interestingly, Bradykinin Receptor B2 (BDKRB2) is also a highly frequent core gene identified in the complement cluster and in the GPCR cluster. Taken together, the regulation of SERPING1 and BDKRB2 may explain the presence of edema and structural changes in the FDMI eye, possibly resulting from the increase in blood vessel permeability during constriction of the choroid via BDKRB2. BDKRB2 has frequently been associated with brain edema, fluid leakage, signaling by GPCR and regulation of actin cytoskeleton and potentially may allow fluids to leak into the retina/vitreous as seen in pathological myopia (*Francisco, Salvador & Amparo, 2015*). Deficiencies in SERPING1 have also previously been shown to occur in age-related macular degeneration (*Ennis et al., 2008*).

The identification of bile acid metabolism as the only statistically significant change during FDMI was not predicted but is compatible with all omics results implicating energy, metabolic and ion transport (*Hysi et al., 2014*; *Kiefer et al., 2013*; *Riddell & Crewther, 2017a*; *Stone & Khurana, 2010*; *Verhoeven et al., 2013*) and ultrastructural fluid movements (*Hysi et al., 2014*; *Kiefer et al., 2013*; *Riddell & Crewther, 2017a*; *Stone & Khurana, 2010*; *Verhoeven et al., 2013*). The finding may be due to the acute time points selected for analysis given that previous ultrastructural descriptions have demonstrated that minimal

change in choroidal thickness occurs prior to 72 h after occluder removal (*Liang et al., 2004*; *Liang et al., 1996*). This finding is not inconsistent with our conceptualisation of FDMR in an eye that has been under physiological stress for 10 days. Recent reports (*Lefebvre et al., 2009*; *Staels & Fonseca, 2009*) also indicate that bile acids regulate not only their own synthesis, but also triglyceride, cholesterol, glucose, and energy homeostasis and play a role in osmoregulation. Bile acid synthesis has also been shown to be inhibited by all-trans retinoic acid by down regulating key bile acid synthesis and metabolism enzymes, such as cytochrome oxidase (CYP7A1, CYP8B1), ion transport (SLC27A5 and AKRLD1) (*Mamoon et al., 2014*; *Yang et al., 2014*) and closely associated with clock genes, metabolism and epigenetic regulators (*Feng & Lazar, 2012*).

The two main limitations of this study are the use of the two types of tissue (retina/RPE and retina/RPE/choroid) and as discussed, the consideration of variance in gene expression between the GSE6543 dataset and GSE89325 dataset due to sample pooling. Our lab has previously assessed the impact of using a combination of ocular tissue in large scale genomic and proteomic studies (*Riddell & Crewther, 2017a*) and found that regardless of the varying combinations of tissues used in studies of myopia, both FD and optical defocus, similar biological mechanisms were identified. Such similarity in identified biological mechanisms suggests that responses to environmental manipulation that reduces focused visual information is to elicit perturbation of the growth response by the whole eye across multiple tissue layers though originating in the photoreceptor layer and regardless of tissue properties and functions. However, tissue type is not a factor for GSEA as the analysis aims to assess combined changes in expression of genes within biological networks. Furthermore, gene analysis of separate ocular tissue compared to combined tissues show differing expression patterns may confer misleading results. For example, the gene BMP2 that *Zhang, Liu & Wildsoet (2012)* reported as mainly localised in the retina of chick, has previously been identified as a potential risk factor for myopia in chick retina/RPE (*McGlinn et al., 2007*) and in chick RPE (*Zhang, Liu & Wildsoet, 2012*). In a further cohort by the same lab (*Zhang et al., 2016*), BMP2 was reported as non-significantly expressed in chick retina. Such contradictory results raise issues about the independence of differential gene signalling by BMP2 suggesting that BMP2 perturbation may be more related to other genes responding to the visual manipulation. In fact, our GSEA analysis that assesses the collective gene expression changes in all genes within all known biological pathways has identified BMP2 as a core gene for the immunological cytokine-cytokine receptor interaction pathway. This suggests that BMP2 is possibly functioning as a modulator for inflammation rather than influencing the growth signal (*He et al., 2018*) in the development of myopia. Thus we contend that pooling RNA from multiple tissues is not an impediment to our GSEA based interpretation as evidenced by the robustness of our current findings with many commonalities between FDMI and FDMR and between the different methodologies and much previous research in human and animals.

## CONCLUSIONS

We believe our analyses demonstrate that GSEA is a valuable tool in identifying altered biological pathways in the chick model of refractive error, as well as providing greater

statistical power in identifying biological pathways not otherwise considered to be of potential significance. A major strength with GSEA is the generation of further hypotheses related to the understanding that many genes within a biological pathway can contribute to the underlying biology of a disease (*Tripathi, Glazko & Emmert-Streib, 2013*). Our findings demonstrate that gene pathway changes in mitochondrial energy metabolism, neurotransmission and subsequent involvement of ion homeostasis are tightly coupled to axial length and refraction changes as early as 6 h and 72 h after application of FD. The suppression in bile acid metabolism during early recovery from profound FDM highlights the importance of maintaining energy metabolism in myopia. The GSEA findings provide supporting evidence for the RIDE model as well as complementing earlier biometric and ultrastructural findings showing that form-deprivation occlusion leads to changes in eye volume, refraction, thinning of the retina and choroid, and morphological evidence for hyperosmolarity (*Brocker, Thompson & Vasiliou, 2012*; *Crewther et al., 2006*; *Grubman et al., 2016*; *Hollborn et al., 2017*; *Junghans et al., 1999*; *Liang et al., 1995*; *Liang et al., 2004*). Although a combination of posterior ocular tissue have been analysed in previous transcriptome studies on refractive errors (*McGlinn et al., 2007*; *Rada & Wiechmann, 2009*; *Riddell et al., 2016*; *Shelton et al., 2008*; *Stone et al., 2011*), commonalities in differentially expressed genes have now been identified regardless of species, tissue analysed and genomic platform (*Riddell & Crewther, 2017a*). Further studies using the GSEA approach may benefit from data collected from next-generation sequencing technologies, as transcripts are sequenced for analysis and genomic annotations are updated. Follow-up studies may also consider refining this analysis in specific retinal cell-types/tissues including the role of the vitreous in ocular development. This study provides an evidence base for further understanding of the biochemical and genetic mechanisms underlying and governing environmentally induced refractive error development in chick with implications for clinical myopia. However, there is need for greater understanding of the effects of FD recovery over a longer period of time, particularly after 72 h where greatest choroidal expansion has been reported (Fig. 1). Future strategies to modify/supplement abnormal mitochondrial dynamics and reduce ionic induction of innate immune responses, may be an attractive therapeutic intervention target.

## ACKNOWLEDGEMENTS

The Magnetic Resonance Image was courtesy of Egan, G., at the Howard Florey Neuroscience Institute.

### Funding

This research was supported by a National Health and Medical Research Council Development grant (ID448606) to David P. Crewther and Sheila G. Crewther and a further Australian Research Council grant (DP110103784). There was no additional external funding received for this study. The funders had no role in study design, data collection and analysis, decision to publish, or preparation of the manuscript.

## Grant Disclosures

The following grant information was disclosed by the authors:
National Health and Medical Research Council Development: ID448606.
Australian Research Council: DP110103784.

## Competing Interests

The authors declare there are no competing interests.

## Author Contributions

- Loretta Giummarra conceived and designed the experiments, performed the experiments, analyzed the data, contributed reagents/materials/analysis tools, prepared figures and/or tables, authored or reviewed drafts of the paper, approved the final draft.
- Sheila G. Crewther conceived and designed the experiments, performed the experiments, contributed reagents/materials/analysis tools, authored or reviewed drafts of the paper, approved the final draft.
- Nina Riddell analyzed the data, contributed reagents/materials/analysis tools, prepared figures and/or tables, authored or reviewed drafts of the paper, approved the final draft.
- Melanie J. Murphy performed the experiments, contributed reagents/materials/analysis tools, authored or reviewed drafts of the paper, approved the final draft.
- David P. Crewther conceived and designed the experiments, contributed reagents/materials/analysis tools, authored or reviewed drafts of the paper, approved the final draft, provided funding.

## Animal Ethics

The following information was supplied relating to ethical approvals (i.e., approving body and any reference numbers):

All animal work in this study was approved by the La Trobe University Animal Ethics Committee (Approval No. 05/07) and is in accordance with the Guidelines for Use of Animals in Research by the National Health and Medical Research Council (NHMRC) of Australia and the ARVO Statement for the Use of Animals in Ophthalmic and Vision Research.

## Microarray Data Availability

The data discussed in this article are available in NCBI's Gene Expression Omnibus under accession number GSE89325.

## Supplemental Information

Supplemental information for this article can be found online at http://dx.doi.org/10.7717/peerj.5048#supplemental-information.

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
