# Peer review of "Pathway analysis identifies altered mitochondrial metabolism, neurotransmission, structural pathways and complement cascade in retina/RPE/ choroid in chick model of form-deprivation myopia"

_PeerJ, doi:10.7717/peerj.5048_

## Round 0.1 · original submission · Major Revisions

As suggested by both reviewers:

Please revise manuscript for clarity, resolve issue related to pooling of samples, results on ocular biometrics and enrichment analysis. Validating core gene expression changes using independent methods and provide rational of the experiments in result section as suggested.

Reviewer 1 ·

Basic reporting

Title: Pathway analysis identifies altered mitochondrial metabolism, neurotransmission, structural pathways and complement cascade in retina/RPE/choroid in chick model of form-deprivation myopia (#24645)

Giummarra L, et al., present an interesting manuscript investigating the mechanisms of form-deprivation myopia and recovery from FD myopia using chick model, and identified the involvement of mitochondrial metabolism, neurotransmission, structural pathways, complement cascade, and bile acid and bile salt metabolism pathways in retina/RPE/choroid. This manuscript is well written and structured. Authors employed genome-wide gene expression profiling and various analysis methods to investigate the role of different signaling pathways in myopia development. This study provides interesting insight into the role of retina/RPE/choroid in FD myopia model, but some questions need to be addressed for the consideration of publication.
1. One of the major concerns for this study is the pooling of samples, or pooling n=5 for each conditions to n=1 for Affymetrix chips. Pooling samples together from a couple of individuals in general is not preferred if researcher can obtain enough sample or RNA from each eye. Collecting enough RNA from each eye should not be a problem for this study since combined retina/RPE/choroid tissues are being collected, and more than 1 ug of RNA can be easily collected from each eye for microarray assay. Authors please give the reason of pooling samples together. Please also give explanation that n=1 for each condition works.

In addition, authors emphasized the similarity between retina/RPE/choroid and retina/RPE (McGlinn et al. study) (Line 153-154), as well as pooled and individual samples (Line 190-197). They also discussed the limitation of this study at the end of discussion. However, the difference between these situations should be discussed more. First, every tissue has unique anatomical properties and functions. For example, choroid is a very complex tissue which include large amount of vessels, endothelium cells and other cell types that are very different from retina and RPE. Retina also contains so many different cell types including neurons that may respond to FD or defocus differently with different period of treatment. While RPE in nature is epithelial cell. These tissues are so different and they may act differently in myopia development. Second, experiments using combined tissues are a kind of practice of pooling sample in nature. Third, pooling samples together from different animals can be problematic in many ways, and should be only considered when no enough sample is available under certain conditions (L190-197).

2. Figure 1 is very confusing from Introduction to Materials and Methods, as well as Results:
Line 69: Is Figure 1 the result of current study or from references? I checked references listed here but did not find anything related to Figure 1. Legend of Figure 1 indicated that it is curtesy of Egan, G. If this is true, then no figure is needed in this manuscript (just add reference), and no need to describe MRI in Materials and Methods. Please revise manuscript to make it clear.
If Figure 1 is the result of this study as described in Materials and Methods – MRI imaging section, then please describe MRI methods and results as appropriate.
Please label Figure 1, including retina, choroid, and sclera in both panels if this is your original research. It is very confusing that authors indicated 24 h of recovery after FD treatment in this study, but MRI figure (1B) was taken 72 h of recovery.

3. Results for ocular biometrics and Figure 2:
It is appreciated that authors collected extra data to compliment McGlinn et al. study, but these data should be plotted separately. The reason is that these data were collected from 7 days old chicks which is different from current study. Plot these data together can be misleading (Figure 2A & 2B).

I suggest plotting Figure 2B AL and VCD in separate bars. The reasons are: 1) it is hard to compare ALs when they are stacked on top of VCDs which ends at different levels. 2) Readers may misinterpreted black + gray bar as the axial length of the eye, and adding AL and VCD in one bar is not meaningful anyway.

4. Enrichment analysis for short-term FDM induction and Table 1
Summarize genes identified from McGlinn et al. study to 13 enriched pathways and Table 1 is not informative. It will be better to provide gene list for each identified pathway.

Minor points include
1. Line 80, line 135, line 156, line 163-164, etc.: Please be consistence with form-deprivation and form deprivation used in manuscript, as well as 6hrs, 72hrs, 6 hr, 24 hr; short term, short-term, long term, long-term;
2. Line 129: Is this study using 10 days FD treatment? Typo here for “following 7 days of translucent occlusion”?
3. Line 187: Typo “n=65”

Experimental design

Authors need to make clear sample number for each experiments

Validity of the findings

no comments

Reviewer 2 ·

Basic reporting

The manuscript entitled “Pathway analysis identifies altered mitochondrial metabolism, neurotransmission, structural pathways and complement cascade in retina/RPE/choroid in chick model of formdeprivation myopia” by Giummarra et. al. has profiled transcriptome wide gene expression changes in during the induction and recovery from form-deprivation myopia (FDM) in chick. The GSEA analysis revealed important pathways in the pathophysiological process. Overall the manuscript is well executed with clear background knowledge and experimental approaches. However, the rationale/hypothesis were not well described in the result section for the experiment/analysis performed. The quality of this manuscript could be improved by addressing the following concerns. At the present stage of the manuscript, I suggest considering for publication in PeerJ following a revision.

Experimental design

1. I suggest validating core gene expression changes, at least one gene of each of the enriched pathways by an independent method (for instance, quantitative PCR method).

Validity of the findings

2. The title of a figure legend should describe the figure succinctly, not the experiment. I suggest using conclusions of the result as title of the result sections as well as Figure legend.
3. I suggest including rational of the experiments, preceding the actual experiment and conclusions in each “Result” sections.
4. Figure1; please move the MRI imaging method to the “Materials and Methods” section.
5. Figure 2 a, b, c, supplementary figure 1; include statistical significance test p and r value on the figures and wherever applicable on other figures.

---

## Round 0.2 · accepted · Accept

The revised manuscript is acceptable in the current form.

# Reviewer 1 ·

Basic reporting

no comment

Experimental design

no comment

Validity of the findings

no comment

Additional comments

no comment

Reviewer 2 ·

Basic reporting

The authors have significantly improved the clarity of the manuscript. I recommend to consider the manuscript for publication in PeerJ after resolving the concerns raised by the first reviewer.

Experimental design

no comment

Validity of the findings

no comment

Additional comments

no comment

---

## Author Rebuttal · Round 0.2

COLLAGE OF SCHIENCE, HEALTH & ENGINEERING
School of Psychology & Public Health, Department of Psychology & Counselling

**Mailing address**
La Trobe University
Victoria 3086 Australia

**T** + 61 3 9479 2290
**E** s.crewther@latrobe.edu.au

**CAMPUSES**
Melbourne (Bundoora)
Albury-Wodonga
Bendigo
City (Collins Street)
Franklin Street (CBD)
Mildura
Shepparton
Sydney

25th April 2018

Shree Ram Singh
Academic Editor
**RE: Pathway Analysis identifies altered mitochondrial metabolism, neurotransmission, structural pathways and complement cascade in retina/RPE/choroid in chick model of form-deprivation myopia**

Dear Dr. Shree Ram Singh

We would like to thank the reviewers for considering our manuscript and for their comments. We have taken note of all suggestions by the reviewers and amended the manuscript to accommodate them wherever appropriate and answered in detail in the Table below (Reviewer Comments in left column and particular Responses in right column) where we think the suggestion blurs our initial aims for the manuscript. We have also endeavoured to improve reader accessibility by adding further information that should better explain the theory behind our methodological decisions.

Please see the following page for the table of reviewer comments and our responses. Thank you for taking the time to consider our manuscript and we look forward to hearing from you

Sincerely,

**Sheila Crewther**
Professor of Neuroscience

ABN 64 804 735 113
CRICOS Provider 00115M

[Figure]

**Reviewer Comments and Authors Responses**

| Reviewer 1 Comments | Author Response |
|---|---|
| Giummarra L, et al., present an interesting manuscript investigating the mechanisms of form-deprivation myopia and recovery from FD myopia using chick model, and identified the involvement of mitochondrial metabolism, neurotransmission, structural pathways, complement cascade, and bile acid and bile salt metabolism pathways in retina/RPE/choroid. This manuscript is well written and structured. Authors employed genome-wide gene expression profiling and various analysis methods to investigate the role of different signaling pathways in myopia development. This study provides interesting insight into the role of retina/RPE/choroid in FD myopia model, but some questions need to be addressed for the consideration of publication. | |
| One of the major concerns for this study is the pooling of samples, or pooling n=5 for each conditions to n=1 for Affymetrix chips. Pooling samples together from a couple of individuals in general is not preferred if researcher can obtain enough sample or RNA from each eye. Collecting enough RNA from each eye should not be a problem for this study since combined retina/RPE/choroid tissues are being collected, and more than 1 ug of RNA can be easily collected from each eye for microarray assay. Authors please give the reason of pooling samples together. Please also give explanation that n=1 for each condition works. | The authors believe there is adequate and robust evidence available to justify the use of pooling especially as our analysis focused on GSEA which assesses the collective changes in gene expression and identifies relevant biological pathways where these genes act. Reference to justification of pooling with regard to GSEA is included in lines 205-216

We have also indicated in lines 227-228 that samples for GSEA are n=1 |
| In addition, authors emphasized the similarity between retina/RPE/choroid and retina/RPE (McGlinn et al. study) (Line 153-154), as well as pooled and individual samples (Line 190-197). They also discussed the limitation of this study at the end of discussion. However, the difference between these situations should be discussed more.
First, every tissue has unique anatomical properties and functions. For example, choroid | We have addressed the reviewer's concerns in lines 619-643 with the following discussion. *"Our lab has previously assessed the impact of using a combination of ocular tissue in large scale genomic and proteomic studies (Riddell & Crewther 2017) and found that regardless of the varying combinations of tissues used in studies of myopia, both FD and optical defocus, similar biological mechanisms were identified. Such similarity in identified biological* |

| | |
|---|---|
| is a very complex tissue which include large amount of vessels, endothelium cells and other cell types that are very different from retina and RPE. Retina also contains so many different cell types including neurons that may respond to FD or defocus differently with different period of treatment. While RPE in nature is epithelial cell. These tissues are so different and they may act differently in myopia development.<br>Second, experiments using combined tissues are a kind of practice of pooling sample in nature.<br>Third, pooling samples together from different animals can be problematic in many ways, and should be only considered when no enough sample is available under certain conditions (L190-197). | *mechanisms suggests that responses to environmental manipulation that reduces focused visual information is to elicit perturbation of the growth response by the whole eye across multiple tissue layers though originating in the photoreceptor layer and regardless of tissue properties and functions. However, tissue type is not a factor for GSEA as the analysis aims to assess combined changes in expression of genes within biological networks. Furthermore, gene analysis of separate ocular tissue compared to combined tissues show differing expression patterns may confer misleading results. For example, the gene BMP2 that Zhang et al. (2012) reported as mainly localised in the retina of chick, has previously been identified as a potential risk factor for myopia in chick retina/RPE (McGlinn et al. 2007) and in chick RPE (Zhang et al. 2012). In a further cohort by the same lab (Zhang et al. 2016), BMP2 was reported as non-significantly expressed in chick retina. Such contradictory results raise issues about the independence of differential gene signalling by BMP2 suggesting that BMP2 perturbation may be more related to other genes responding to the visual manipulation. In fact, our GSEA analysis that assesses the collective gene expression changes in all genes within all known biological pathways has identified BMP2 as a core gene for the immunological cytokine-cytokine receptor interaction pathway. This suggests that BMP2 is possibly functioning as a modulator for inflammation rather than influencing the growth signal (He et al. 2018) in the development of myopia. Thus we contend that pooling RNA from multiple tissues is not an impediment to our GSEA based interpretation as evidenced by the robustness of our current findings with many commonalities between FDMI and FDMR and between the different methodologies and much previous research in human and animals."* |
| Second, experiments using combined tissues | As discussed above, we believe pooling has not |

| | |
|---|---|
| are a kind of practice of pooling sample in nature. | impacted on our pathway enrichment results. Please refer to lines 205-216 |
| Figure 1 is very confusing from Introduction to Materials and Methods, as well as Results: Line 69: Is Figure 1 the result of current study or from references? I checked references listed here but did not find anything related to Figure 1. Legend of Figure 1 indicated that it is curtesy of Egan, G. If this is true, then no figure is needed in this manuscript (just add reference), and no need to describe MRI in Materials and Methods. Please revise manuscript to make it clear.<br><br>If Figure 1 is the result of this study as described in Materials and Methods – MRI imaging section, then please describe MRI methods and results as appropriate.<br><br>Please label Figure 1, including retina, choroid, and sclera in both panels if this is your original research. It is very confusing that authors indicated 24 h of recovery after FD treatment in this study, but MRI figure (1B) was taken 72 h of recovery. | The aim of Fig 1 is to confirm the retinal thinning and choroidal expansion observations previously identified histologically by our lab (Liang et al. 2004). Hence we selected 72h post FD-recovery as this timepoint identified the greatest changes in these tissues. We have also described Figure 1 in the results section of the manuscript in lines 308-310.<br><br><br><br>We have expanded the methods to include detail of the MRI imaging and have also included labels in Fig 1.<br><br><br>As requested, we have included labels in figure 1 (retina, choroid and sclera). |
| Results for ocular biometrics and Figure 2: It is appreciated that authors collected extra data to compliment McGlinn et al. study, but these data should be plotted separately. The reason is that these data were collected from 7 days old chicks which is different from current study. Plot these data together can be misleading (Figure 2A & 2B). | As requested, the figure has been revised to show these data separately. Biometrics to accompany McGlinn et al time-points are shown in Fig 2a and 2b. Biometics for the FDMR time-points are shown in Fig 2c and 2d |
| I suggest plotting Figure 2B AL and VCD in separate bars. The reasons are: 1) it is hard to compare ALs when they are stacked on top of VCDs which ends at different levels. 2) Readers may misinterpreted black + gray bar as the axial length of the eye, and adding AL and VCD in one bar is not meaningful anyway. | As requested, the figure has been revised to show AL and VCD as separate bars |
| Enrichment analysis for short-term FDM induction and Table 1 Summarize genes identified from McGlinn et al. study to 13 enriched pathways and Table 1 is | Highly enriched genes for each pathway in Table 1 have now been included |

| | |
|---|---|
| not informative. It will be better to provide gene list for each identified pathway. | |
| Line 80, line 135, line 156, line 163-164, etc.: Please be consistence with form-deprivation and form deprivation used in manuscript, as well as 6hrs, 72hrs, 6 hr, 24 hr; short term, short-term, long term, long-term; | These terms have been amended to 'form-deprivation", "hr", "short-term", and "long-term" |
| Line 129: Is this study using 10 days FD treatment? Typo here for "following 7 days of translucent occlusion"? | Amended. Line 129 is now line 131 |
| Line 187: Typo "n=65" | Amended. Line 187 is now line 196 |
| Authors need to make clear sample number for each experiments | We have aimed to clarify this in lines 227-228 |

| Reviewer 2 Comments | Authors Response |
|---|---|
| The manuscript entitled "Pathway analysis identifies altered mitochondrial metabolism, neurotransmission, structural pathways and complement cascade in retina/RPE/choroid in chick model of form deprivation myopia" by Giummarra et. al. has profiled transcriptome wide gene expression changes in during the induction and recovery from form-deprivation myopia (FDM) in chick. The GSEA analysis revealed important pathways in the pathophysiological process. Overall the manuscript is well executed with clear background knowledge and experimental approaches. However, the rationale/hypothesis were not well described in the result section for the experiment/analysis performed. The quality of this manuscript could be improved by addressing the following concerns. At the present stage of the manuscript, I suggest considering for publication in PeerJ following a revision. | |
| I suggest validating core gene expression changes, at least one gene of each of the enriched pathways by an independent method (for instance, quantitative PCR method). | The microarray results described here are consistent with our previously published work using RNA-seq (Riddell et al. 2016). Hence we chose not to validate core genes from each pathway by qPCR or other molecular technique |

| | |
|---|---|
| | as there are >130 pathways identified in this study. Furthermore, validation of the core genes using qPCR is also often questionable as it is reportedly subject to within-lab and technical differences (microarray vs qPCR) (Nygaard & Hovig 2009). Lastly, microarrays have been shown to exhibit good sensitivity and specificity in detecting gene expression changes (Dago et al. 2014) and perform comparatively to RNA-seq, as indicated above in our lab particularly (Riddell et al. 2016)<br><br>Additionally, single-gene analysis will not confirm significance of biological pathways identified in GSEA. For example, In the FDMI dataset, EIF4EBP1 was highly ranked at the top of the gene list for GSEA (ie. Highly upregulated; see attached supplementary information). This gene was only identified as a core gene in the Translation pathway. A gene that was highly ranked at the bottom of the list (ie highly down regulated) was GDAP1. Interestingly, this gene was not listed as a core gene for any of the significant pathways identified by GSEA. This suggests that highly regulated genes are not always responsible for driving treatment-specific biological responses.<br><br>This discussion has been included in the manuscript at lines 486-500 |
| The title of a figure legend should describe the figure succinctly, not the experiment. I suggest using conclusions of the result as title of the result sections as well as Figure legend. | The authors believe that the figure titles accurately describe the figure. Due to the complexity of the data, the authors believe that it is not ideal to have conclusions of the results as the figure titles as the conclusions presented in the results is drawn from more than one figure. |
| I suggest including rational of the experiments, preceding the actual experiment and conclusions in each "Result" sections | The authors have included a rationale preceding each experiment in the results section of the manuscript in lines 315-316, 326-328, 340-343, 368-372, 468-469 |
| Figure1; please move the MRI imaging method to the "Materials and Methods" section. | We have moved the MRI imaging method to the "Materials and Methods section as requested |

[Figure]

| | |
|---|---|
| Figure 2 a, b, c, supplementary figure 1; include statistical significance test p and r value on the figures and wherever applicable on other figures. | Statistical analysis have been provided in the manuscript (lines 288 – 306). We have chosen not to indicate significance in Fig 2 to maintain image clarity. However to illustrate this, we have provided a version of Fig 2 (supplementary figure 2) with statistical significance indicated. The authors believe that statistical significance of the biometric data is not the main aim of the paper and had no effect on the interpretation of the GSEA results. |

# References

Dago D, Malerba G, Ferarrini A, and Delledonne M. 2014. Evaluation of microarray sensitivity and specificity in gene expression differential analysis by RNA-seq and quantitative RT-PCR. *Journal of Multidisciplinary Scientific Research* 20142:5-9.

He L, Frost MR, Siegwart JT, Jr., and Norton TT. 2018. Altered gene expression in tree shrew retina and retinal pigment epithelium produced by short periods of minus-lens wear. *Exp Eye Res* 168:77-88. 10.1016/j.exer.2018.01.005

Liang H, Crewther SG, Crewther DP, and Junghans BM. 2004. Structural and elemental evidence for edema in the retina, retinal pigment epithelium, and choroid during recovery from experimentally induced myopia. *Investigative Ophthalmology & Visual Science* 45:2463-2474.

McGlinn AM, Baldwin DA, Tobias JW, Budak MT, Khurana TS, and Stone RA. 2007. Form-deprivation myopia in chick induces limited changes in retinal gene expression. *Invest Ophthalmol Vis Sci* 48:3430-3436. 10.1167/iovs.06-1538

Nygaard V, and Hovig E. 2009. Methods for quantitation of gene expression. *Front Biosci (Landmark Ed)* 14:552-569.

Riddell N, and Crewther SG. 2017. Integrated Comparison of GWAS, Transcriptome, and Proteomics Studies Highlights Similarities in the Biological Basis of Animal and Human Myopia. *Invest Ophthalmol Vis Sci* 58:660-669. 10.1167/iovs.16-20618

Riddell N, Giummarra L, Hall NE, and Crewther SG. 2016. Bidirectional Expression of Metabolic, Structural, and Immune Pathways in Early Myopia and Hyperopia. *Front Neurosci* 10:390. 10.3389/fnins.2016.00390

Zhang Y, Liu Y, Hang A, Phan E, and Wildsoet CF. 2016. Differential gene expression of BMP2 and BMP receptors in chick retina & choroid induced by imposed optical defocus. *Vis Neurosci* 33:E015. 10.1017/S0952523816000122

Zhang Y, Liu Y, and Wildsoet CF. 2012. Bidirectional, optical sign-dependent regulation of BMP2 gene expression in chick retinal pigment epithelium. *Invest Ophthalmol Vis Sci* 53:6072-6080. 10.1167/iovs.12-9917

[Figure]

COLLAGE OF SCHIENCE, HEALTH & ENGINEERING
School of Psychology & Public Health, Department of Psychology & Counselling